# Preserved stem cell content and innervation profile of elderly human skeletal muscle with lifelong recreational exercise

Casper Soendenbroe[1,2,3] (ID), Christopher L. Dahl[1], Christopher Meulengracht[1], Michal Tamáš[1] (ID), Rene B. Svensson[1,3] (ID), Peter Schjerling[1,3] (ID), Michael Kjaer[1,3] (ID), Jesper L. Andersen[1,3] and Abigail L. Mackey[1,2,3] (ID)

[1]*Institute of Sports Medicine Copenhagen, Department of Orthopedic Surgery M, Copenhagen University Hospital – Bispebjerg and Frederiksberg, Copenhagen, Denmark*
[2]*Xlab, Department of Biomedical Sciences, Faculty of Health and Medical Sciences, University of Copenhagen, Copenhagen, Denmark*
[3]*Center for Healthy Aging, Department of Clinical Medicine, Faculty of Health and Medical Sciences, University of Copenhagen, Copenhagen, Denmark*

Edited by: Richard Carson & Kevin Murach

Linked articles: This article is highlighted in a Perspective article by Englund and a Journal Club article by O'Bryan & Hiam. To read these articles, visit https://doi.org/10.1113/JP283015 and https://doi.org/10.1113/JP283102.

The peer review history is available in the Supporting Information section of this article (https://doi.org/10.1113/JP282677#support-information-section).

**The Journal of Physiology**

**Abstract** Muscle fibre denervation and declining numbers of muscle stem (satellite) cells are defining characteristics of ageing skeletal muscle. The aim of this study was to investigate the potential for lifelong recreational exercise to offset muscle fibre denervation and compromised

**Casper Søndenbroe** received his bachelor and master's degree in Sport Science and Physiology at the Faculty of Science at University of Copenhagen, Denmark. His PhD training was undertaken at the Institute of Sports Medicine Copenhagen at Bispebjerg Hospital and at the Centre for Healthy Ageing at the Faculty of Health and Medical Science, University of Copenhagen, Denmark, where he was supervised by Associate Professor Abigail L. Mackey. The research is focused on the neuromuscular system and how it is affected by ageing and exercise, with a strong emphasis on human studies.

satellite cell content and function, both at rest and under challenged conditions. Sixteen elderly life-long recreational exercisers (LLEX) were studied alongside groups of age-matched sedentary (SED) and young subjects. Lean body mass and maximal voluntary contraction were assessed, and a strength training bout was performed. From muscle biopsies, tissue and primary myogenic cell cultures were analysed by immunofluorescence and RT-qPCR to assess myofibre denervation and satellite cell quantity and function. LLEX demonstrated superior muscle function under challenged conditions. When compared with SED, the muscle of LLEX was found to contain a greater content of satellite cells associated with type II myofibres specifically, along with higher mRNA levels of the beta and gamma acetylcholine receptors (AChR). No difference was observed between LLEX and SED for the proportion of denervated fibres or satellite cell function, as assessed *in vitro* by myogenic cell differentiation and fusion index assays. When compared with inactive counterparts, the skeletal muscle of lifelong exercisers is characterised by greater fatigue resistance under challenged conditions *in vivo*, together with a more youthful tissue satellite cell and AChR profile. Our data suggest a little recreational level exercise goes a long way in protecting against the emergence of classic phenotypic traits associated with the aged muscle.

(Received 2 December 2021; accepted after revision 14 February 2022; first published online 28 February 2022)

**Corresponding authors** Casper Soendenbroe and Abigail L. Mackey: Building 8, 1st floor, Nielsine Nielsens Vej 11, Copenhagen, Denmark 2400, Denmark.    Emails: Caspersoendenbroe@outlook.dk; abigailmac@sund.ku.dk

**Abstract figure legend** Lifelong exercisers were studied alongside age-matched sedentary individuals and young subjects. Muscle biopsies were obtained from all subjects and used for immunofluorescent analyses and cell culture experiments. *In vivo* measurements of muscle mass and function were also performed. Lifelong exercise was associated with a preserved number of type II myofibre-associated satellite cells, an improved innervation status that was similar to the young control group, and better muscle function under challenged conditions. The findings suggest that even low amounts of physical activity over many years have a positive impact on muscle health and innervation status. Figure was created using BioRender. Publication licence has been obtained.

## Key points

- The detrimental effects of ageing can be partially offset by lifelong self-organized recreational exercise, as evidence by preserved type II myofibre-associated satellite cells, a beneficial muscle innervation status and greater fatigue resistance under challenged conditions.
- Satellite cell function (*in vitro*), muscle fibre size and muscle fibre denervation determined by immunofluorescence were not affected by recreational exercise.
- Individuals that are recreationally active are far more abundant than master athletes, which sharply increases the translational perspective of the present study. Future studies should further investigate recreational activity in relation to muscle health, while also including female participants.

## Introduction

Age-related loss of muscle mass and function is often unnoticeable and negligible during mid-life, but gradually accelerates, causing most individuals entering their eighth decade of life to have a greatly diminished muscle function (Janssen *et al.* 2000; Kostka, 2005; Suetta *et al.* 2019). Among the myriad changes associated with the ageing muscle, myofibre denervation and a decline in the number (Verdijk *et al.* 2014) and function (Pietrangelo *et al.* 2009) of muscle stem (satellite) cells are clear features. Myofibre denervation occurs following decay of $\alpha$-motoneurons in the spinal cord (Campbell *et al.* 1973; Tomlinson & Irving, 1977; Mittal & Logmani, 1987; Power *et al.* 2010; Piasecki *et al.* 2016) or destabilization of neuromuscular junctions (NMJ) (Bütikofer *et al.* 2011). Loss of myofibre innervation removes the transcriptional specialization normally confined to the small synaptic area, and alters gene expression in the extra-synaptic area of the myofibre (Covault & Sanes, 1985). For example, a strong upregulation of the acetylcholine receptors (AChR), normally confined to the NMJ, is evident along the length of the myofibre upon denervation (Merlie *et al.* 1984). We (Karlsen *et al.* 2019, 2020; Soendenbroe *et al.* 2019, 2020) and others (Gigliotti *et al.* 2015; Baehr *et al.* 2016; Kelly *et al.* 2018; Sonjak *et al.* 2019; Daou *et al.* 2020; Skoglund *et al.* 2020; Lagerwaard *et al.* 2021; Monti *et al.* 2021) have availed ourselves of this to indirectly investigate myofibre innervation status in human muscle tissue.

Satellite cells are indispensable during embryonic myogenesis and for muscle regeneration during adulthood (Engquist & Zammit, 2021), due to their ability to proliferate, fuse and form myotubes. Given their role as the sole source of myonuclei, satellite cells are also involved in the hypertrophic response to exercise (Murach *et al.* 2021*a*). Studies using satellite cell-depleted mice have shown that some hypertrophy can be achieved without satellite cells, but in order to maximize the response to long-term training, satellite cells are required (Englund *et al.* 2020). It is now also clear that satellite cells interact directly with muscle fibres (Murach *et al.* 2021*b*) and with other cell types located in the microenvironment surrounding the muscle fibre, including fibroblasts (Fry *et al.* 2017; Mackey *et al.* 2017) and endothelial cells (Nederveen *et al.* 2021). Maladaptation of the muscle is evident during persistent overload in the absence of satellite cells, such as increased extracellular matrix and fibroblast number, indicating a regulatory role for satellite cells in ameliorating unfavourable remodelling of the muscle environment (Murach *et al.* 2018). In relation to the NMJ, it has been shown that a subgroup of satellite cells generate and maintain the specialized myonuclei at the NMJ (Liu *et al.* 2017; Larouche *et al.* 2021) and that depletion of satellite cells dampens the regeneration of NMJs following nerve damage (Liu *et al.* 2015). Although not completely depleted, the aged human muscle has been shown to have fewer satellite cells, especially those associated with type II fibres (Verdijk *et al.* 2007, 2014; Karlsen *et al.* 2019, 2020). Furthermore, a link between denervation and satellite cells has been shown, where satellite cells exit the quiescent state following denervation and mount an attempt at compensatory myogenesis (Borisov *et al.* 2001). Long-term denervated fibres also possess viable satellite cells with preserved renewal capability (Wong *et al.* 2021).

A key tool in improving muscle function is increasing levels of physical activity (Pahor *et al.* 2020). Numerous studies have documented the beneficial effects of intense, supervised and short-term interventions (<1 year) on muscle mass (Gylling *et al.* 2020), strength (Erskine *et al.* 2011) and other parameters of health (Nordby *et al.* 2012). However, while short-term interventions of increased physical activity undoubtedly remain an effective countermeasure against age-related loss of muscle function, the effects of self-organized physical activity are less clear. Most studies on aged exercising and sedentary individuals focus on aged master athletes, meaning the best-functioning individuals within their age group (Harridge & Lazarus, 2017), which is a highly select group that constitutes a minor proportion of the general population (Ng & Popkin, 2012). Less than 20% of men and women aged ≥60 performed ≥20 min of vigorous intensity physical activity on three or more days per week (Hallal *et al.* 2012). In contrast, the group of recreationally active individuals constituted around 60%. From the master athlete studies we know that high levels of physical activity, maintained over many years, preserve muscle mass, strength and power (Klitgaard *et al.* 1990; Grassi *et al.* 1991; Mikkelsen *et al.* 2013; Mosole *et al.* 2014). Furthermore, electrophysiological (Power *et al.* 2010) and muscle biopsy (Mosole *et al.* 2014; Sonjak *et al.* 2019) studies indicate that exercise influences the neuromuscular system, possibly by facilitating myofibre reinnervation. However, there exists a paucity of knowledge on recreationally active individuals, especially in relation to myofibre morphology, satellite cell numbers and function, and how these relate to indices of muscle denervation.

The potential of exercise to influence the neuromuscular system is substantial. However, there are discrepancies in outcomes between experimental and self-organized exercise interventions, as well as limited data on recreationally active individuals compared with master athletes. We therefore designed the present study to investigate muscle morphology, satellite cells and myofibre denervation in two well-matched groups of elderly individuals different only in their physical activity history. We hypothesized that physically active individuals would possess a higher lean body mass and better muscle function than sedentary individuals, although an inherent decline in muscle morphology and function due to ageing would still exist (relative to the young control group). Furthermore, we hypothesized that positive effects of lifelong recreational physical activity would be evident for indices of myofibre denervation, myofibre size, type II myofibre-associated satellite cells, and satellite cell function in cell culture in comparison with a sedentary lifestyle.

## Methods

### Ethical approval and participants

Experimental procedures were approved by The Committees on Health Research Ethics for The Capital Region of Denmark (Ref: H-19000881) and were conducted according to the standards set by the *Declaration of Helsinki*, except for registration in a database. Participants signed an informed consent agreement. Two hundred and twenty-three men responded to either newspaper or online advertisements and were screened by telephone and asked wide-ranging questions on their physical activity pattern. Exclusion criteria were age between 40 and 67, obesity (body mass index (BMI) >32 kg/m$^2$), smoking, >14 alcoholic beverages per week, prior muscle biopsies (vastus lateralis), knee pain, current disease and use of anti-coagulant medication.

Fifty-six men were included into one of three groups: young, elderly lifelong exercise (LLEX) and elderly sedentary (SED). Seven individuals did not complete the study: injury not related to study (1), loss of interest (1), knee pain during exercise protocol (1), muscle biopsy only obtained from one leg (3) or no information (1). Subjects in the LLEX group correspond to Tier 1 in the participant classification framework by McKay *et al.* (2022). These individuals meet the recommendations for physical activity set by the World Health Organization, often through a combination of different activities, and without a specific aim at competing. Three additional LLEX subjects were excluded, as they ultimately proved markedly less trained in comparison with the rest of the group. The final number of participants included was 46 (15 young, 16 LLEX and 15 SED).

Young and SED were healthy and had not performed structured physical activity, such as regular football or resistance exercise, or any physical activity during everyday life (e.g. cycling or walking for transportation) for at least 10 (young) or 30 (SED) years prior to enrolment. LLEX had performed multiple sports throughout their adult life. We sought to include participants who had at least partially performed sports which would lead to recruitment of type II myofibres in the lower extremities (high force or high speed). Specific activities reported were as follows (individuals performing each activity; individuals performing each activity as their primary activity): strength training (10;3), ball games (5;3), racket sports (5;3), cycling (5;3), rowing (4;1), running (4;1), gymnastics (3;1), athletics (2;1), martial arts (1;0) and swimming (1;0).

### Study design

The study was comprised of three visits to the research facility, taking place between 08.00 and 13.00 (Fig. 1*A*).

The participants were instructed to refrain from physical activity from two days before visit 1 and for the entire course of study, and they were asked to transport themselves to the institute by car or public transportation. On visits 1 and 3 they were instructed to drink a provided protein shake (Bodylab ShakeUp!, 330 ml, 26 g protein, 284 kcal) at home 2 h before the experiment started instead of their normal breakfast.

Visit 1 consisted of a dual energy x-ray absorptiometry (DEXA) scan, blood sampling, maximal strength testing and a bout of unilateral heavy resistance exercise. Visit 2 consisted of a blood sample. On visit 3, another blood sample was taken, followed by bilateral muscle biopsies.

The leg that was subjected to the exercise bout was block-randomized for dominant/non-dominant, resulting in 8/7 (young), 8/8 (LLEX) and 6/9 (SED). The SED group ended up being unbalanced, as two participants dropped out after being allocated to a group.

### DEXA scan

Thirty minutes before the scan, the participants drank 0.5 l of water, and they emptied their bladder immediately before lying down in the scanner (Lunar DPX-IQ, GE-Healthcare). The participants were carefully positioned and lay supine for 10 min before the scan. Lean body mass (LBM), total bone mineral content, fat percentage and android fat mass were chosen as the outcomes.

### Blood samples

Blood samples were obtained from an antecubital vein. General health parameters were analysed on visit 1, and creatine kinase was analysed on all visits, following standard methods at the Department of Clinical Biochemistry.

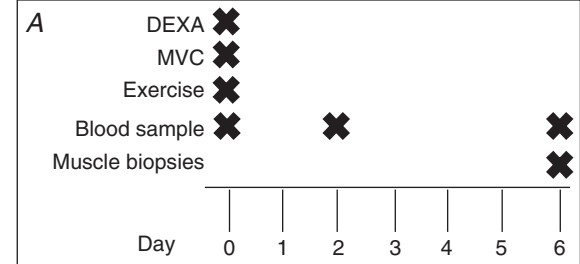

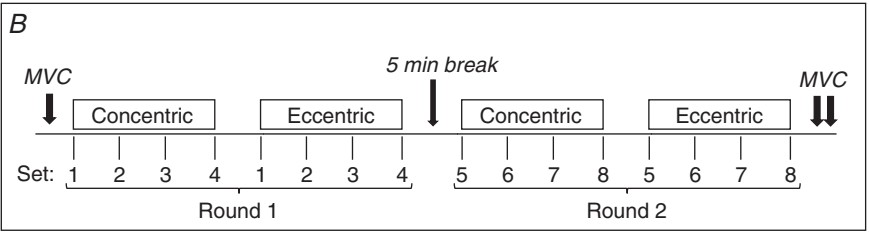

**Figure 1. Study design and exercise protocol**
*A*, three visits spread over 7 days, with timing of exercise, blood samples and biopsies indicated. *B*, unilateral bout of heavy resistance exercise performed on visit 1. Two rounds, separated by a 5–10 min break, each consisting of four sets of concentric and four eccentric isokinetic contractions. The first, fifth and 10th concentric repetitions and the first, third and fifth eccentric repetition from each set was sampled. Maximal voluntary contractions were performed before and immediately after the exercise bout and after a 5 min break. Abbreviations: DEXA, dual energy x-ray absorptiometry; MVC, maximal voluntary contraction.

## Maximal voluntary contraction

Participants had their assigned leg tested for maximal voluntary contraction (MVC) in a dynamometer (KinCom, model 500−11; Kinetic Communicator). The protocol was similar to the one used in our previous study (Karlsen *et al.* 2020), except angular velocity was 30°/s (2.67 s per repetition). The isometric test was repeated after the exercise bout.

## Acute resistance exercise bout

Participants underwent a bout of unilateral heavy resistance exercise in the KinCom using the same leg as for the MVC. The exercise protocol is illustrated in Fig. 1*B*. Two rounds were performed separated by a 5−10 min break. Each round consisted of four sets of 10 concentric contractions (30°/s) at >70% of MVC. This was followed by four sets of five eccentric contractions (30°/s) at >100% of MVC. Torque was sampled from the first, middle and last repetition from each set. Verbal encouragement and visual feedback were provided. The participants rested for 1.5−2.5 min between sets.

## Muscle biopsy

Muscle biopsies were obtained from the middle portion of the vastus lateralis muscle from both legs. Biopsies were taken under local anaesthetic (1% lidocaine), using the percutaneous needle biopsy technique (Bergstrom, 1975) with manual suction. Care was taken to align the incision sites between the legs. Two biopsies were taken from each leg in immediate succession, through the same incision, with the biopsy needle angled proximally and distally from the incision. Pieces of muscle appropriate for histology were carefully aligned in Tissue-Tek (Sakura Finetek), frozen in liquid nitrogen-cooled isopentane (JT Baker) and stored at -80°C. The remaining tissue was immediately processed for cell culture.

## Cell culture

The cell culture protocol has previously been described in detail (Agley *et al.* 2017; Bechshøft *et al.* 2019). Briefly, tissue was digested using collagenase B (11088815001; Roche) and dispase II (D4693; Sigma-Aldrich) for 1 h in a humidified incubator (37°C and 5% $CO_2$), then filtered through a cell strainer (352340; BD Falcon) and transferred to a cell culture flask (690170/658170; Cellstar) and grown in culture medium (C-23060; PromoCell) until ∼80% confluency (mean 6.3 ± 1.4 SD days). The medium was changed after 3 days and old medium was spun down, and unattached cells were returned to the flask. Afterwards, the medium was changed every second day. Cells were detached using diluted Trypsin-EDTA (25200-056; Gibco) and then incubated with MACS running buffer (130-091-221; Miltenyi Biotec) and CD56 magnetic beads (130-050-401; Miltenyi Biotec). Cells were passed through a pre-separation filter (130-041-407; Miltenyi Biotec) and a large cell column (130-042-202; Miltenyi Biotec) attached to a MultiStand magnet (130-090-312; Miltenyi Biotec), capturing the CD56$^+$ (myogenic) fraction. Approximately 3000 and 5000 CD56$^+$ cells/cm$^2$ were plated on glass coverslips (0111580; Marienfeld) in 12-well plates (353503; Corning), for proliferation (PRO) and differentiation (DIF) experiments. Three 12-well plates were used for PRO and DIF each, and cells were plated in duplicate (IHC or RNA) on each plate, providing three replicates for each analysis. Control leg and exercised leg for each participant were cultured on the same plates. Cells were cultured for 3 days for PRO and 3 + 4 days for DIF. After 3 days in CM, PRO cells were exposed to 10 $\mu$M of 5-bromo-2-deoxyuridine (BrdU) for 5 h. For DIF, the cells were also cultured in CM for 3 days, after which the medium was changed to differentiation medium (C-23260; PromoCell). The medium was changed again after 2 days, and the experiment was stopped after further 2 days. At the end of PRO and DIF, the cells were either fixed using Histofix (Histolab) for immunostaining or processed for RNA extraction.

## RNA extraction

Coverslips containing cells were moved to an empty well in a new plate. One millilitre of TriReagent (TR118; Molecular Research Inc.) was added and, after pipetting several times, the mixture was moved to a 2 ml BioSpec tube (5225; Bio Spec Products Inc.) and stored in a -80°C freezer. At the end of the experiment, all samples were thawed, and RNA purified with added glycogen as previously described (Bechshøft *et al.* 2019).

For the tissue samples, 100 sections (10 $\mu$m each) from the frozen biopsies were transferred to the 2 ml BioSpec tubes and dissolved in 1 ml TriReagent by shaking with five steel beads (2.3 mm, BioSpec) for 15 s in a FastPrep homogenizer (MP Biomedicals). The RNA was purified as for the cell culture, except no glycogen was added.

## Real-time RT-qPCR

Fifty nanograms (cell culture) or 400 ng (tissue) total RNA per sample was converted to cDNA using OmniScript reverse transcriptase (Qiagen) and poly-dT (Qiagen) as previously described (Bechshøft *et al.* 2019). 0.25 $\mu$l cDNA was amplified in a 25 $\mu$l SYBR green polymerase chain reaction (PCR) containing 1×Quantitect SYBR Green Master Mix (Qiagen) and 100 nм of each primer

**Table 1. Primers used for PCR**

| mRNA | Gene name | Genbank | Sense | Antisense |
|------|-----------|---------|-------|-----------|
| RPLP0 | RPLP0 | NM_053275.3 | GGAAACTCTGCATTCTCGCTTCCT | CCAGGACTCGTTTGTACCCGTTG |
| GAPDH | GAPDH | NM_002046.4 | CCTCCTGCACCACCAACTGCTT | GAGGGGCCATCCACAGTCTTCT |
| AChRα1 | CHRNA1 | NM_000079.3 | GCAGAGACCATGAAGTCAGACCAGGAG | CCGATGATGCAAACAAGCATGAA |
| AChRβ1 | CHRNB1 | NM_000747.2 | TTCATCCGGAAGCCGCCAAG | CCGCAGATCAGGGGCAGACA |
| AChRδ | CHRND | NM_000751.2 | CAGCTGTGGATGGGGCAAAC | GCCACTCGGTTCCAGCTGTCTT |
| AChRε | CHRNE | NM_000080.4 | TGGCAGAACTGTTCGCTTATTTTCC | TTGATGGTCTTGCCGTCGTTGT |
| AChRγ | CHRNG | NM_005199.4 | GCCTGCAACCTCATTGCCTGT | ACTCGGCCCACCAGGAACCAC |
| MuSK | MUSK | NM_005592.3 | TCATGGCAGAATTTGACAACCCTAAC | GGCTTCCCGACAGCACACAC |
| MyHCe | MYH3 | NM_002470.3 | CGGATATCGCAGAATCTCAAGTCAA | CTCCAGAAGGGCTGGCTCACTC |
| MyHCn | MYH8 | NM_002472.2 | CGGAAACATGAGCGACGAGTAAAA | CAGCCTGAGAACATTCTTGCGATCTT |
| COL1A1 | COL1A1 | NM_000088.3 | GGCAACAGCCGCTTCACCTAC | GCGGGGAGGTCTTGGTGGTTTT |
| Myogenin | MYOG | NM_002479.5 | CTGCAGTCCAGAGTGGGGCAGT | CTGTAGGGTCAGCCGTGAGCAG |
| p16 | CDKN2A | NM_000077.4 | GGGGGCACCAGAGGCAGTAA | TTCTCAGAGCCTCTCTGGTTCTTTCA |

Abbreviations: RPLP0, Ribosomal Protein Large P0; GAPDH, Glyceraldehyde-3-Phosphate Dehydrogenase; AChR, acetylcholine receptor; MuSK, muscle-specific-kinase; MyHCn, neonatal myosin heavy chain; MyHCe, embryonic myosin heavy chain.

for every target mRNA (Table 1). An MX3005P real-time PCR machine (StrataGene) was used for monitoring the amplification, and a standard curve was made with known concentrations of DNA oligonucleotides (Ultramer oligos, Integrated DNA Technologies) corresponding to the expected PCR product. The Ct values were related to the standard curve. Melting curve analysis after amplification was used to confirm the specificity of the PCR products, and RPLP0 mRNA was originally chosen as the internal control for normalization. To support the use of RPLP0, another unrelated 'constitutive' mRNA, GAPDH, was measured (normalized to RPLP0) and showed no change in response to exercise (shown together with the rest of the mRNA). But the basal level was higher in the young group showing that either GAPDH mRNA decrease by age or that RPLP0 mRNA increase by age. As the first would suggest lower metabolic activity in aged muscle and the latter more protein synthesis, we find the first more likely and therefore used RPLP0 as normalizer for all the mRNA. The data are expressed relative to the SED group (control leg) or for the exercised leg relative to the individual control leg (exercise response).

## Immunofluorescence

Biopsies of both legs from each participant were sectioned (10 $\mu$m) using a cryostat, placed in duplicate on the same glass slide, and stored at -80°C. Four serial sections were used (Table 2): slide 1, dystrophin+MyHCn; slide 2, dystrophin+myosin I (A4.951); slide 3, dystrophin+CD56 (NCAM); slide 4, merosin+phalloidin+desmin (Fig. 2). Additionally, a fifth consecutive slide from selected samples suspected to contain myotendinous junction (MTJ), were stained

for collagen 22 (Koch *et al.* 2004). Satellite cells were stained with Pax7, laminin and myosin I (BA.D5). The slides for Pax7 staining were fixed using Histofix before incubation with the primary antibodies. All other stainings were fixed after incubation with secondary antibodies. Sections were incubated overnight at 5°C with primary antibodies diluted in blocking buffer consisting of 1% BSA and 0.1% sodium azide in Tris-buffered saline (TBS). Then, slides were incubated for 45 min at room temperature with secondary antibodies diluted in blocking buffer. Slides were washed in TBS between each step. Sections were finally mounted with cover glasses using Prolong-Gold-Antifade (P36931; Thermo Fisher Scientific) containing 4′,6-diamidino-2-phenylindole (DAPI).

The immunofluorescence staining protocol for the cultured cells has been described before (Bechshøft *et al.* 2019). Briefly, cells were tritonized (9002-93-1; Sigma-Aldrich) for 8 min and incubated overnight with primary antibodies (desmin and myogenin for DIF and desmin and BrdU for PRO) diluted in blocking buffer (1% BSA and 0.1% sodium azide in TBS). Cells were incubated for 1 h at room temperature with secondary antibodies diluted in blocking buffer. Coverslips containing the cells were mounted on glass slides using Prolong-Gold-Antifade containing DAPI.

## Microscopy

Tissue biopsy sections were imaged using a 20×/0.50 NA (slide 4) or a 10×/0.30 NA objective and a 0.5× camera (DP71, Olympus) mounted on a BX51 Olympus microscope. Greyscale 4080 × 3072 or 2040 × 1513 pixel images were obtained, and sections stained with MyHCn

**Table 2. Primary and secondary antibodies used for immunofluorescence microscopy**

| Primary antibody | | | | |
|---|---|---|---|---|
| Host | Antibody | Company | Cat. no. | Concentration |
| Rabbit | Laminin | Dako | Z0097 | 1:500 |
| Rabbit | Desmin, IgG | Abcam | AB32362 | 1:500–1:1000 |
| Mouse | Dystrophin, IgG2b | Sigma-Aldrich | D8168 | 1:500 |
| Mouse | Myosin 1, IgG1 | DSHB | A4.951 | 1:200 |
| Mouse | Pax 7, IgG1 | DSHB | PAX7 | 1:100 |
| Mouse | Myosin 1, IgG2b | DSHB | BA.D5 | 1:100 |
| Mouse | Merosin Laminin $\alpha$2 | Leica | MEROSIN-CE | 1:200 |
| Mouse | MyHCn, IgG1 | Novocastra | NCL-MHCn | 1:100 |
| Mouse | CD56 (NCAM), IgG1 | Becton Dickinson | 347740 | 1:50 |
| Mouse | Myogenin, IgG1 | DSHB | F5D-s | 1:50 |
| Guinea pig | Collagen 22 | * | KG36 | 1:5000 |
| | Phalloidin 680 | Invitrogen | A22286 | 1:40 |

| Secondary antibody | | | | |
|---|---|---|---|---|
| Host | Antibody | Company | Cat. no. | Concentration |
| Goat | Anti-Mouse 488, IgG | Invitrogen | A-11029 | 1:500 |
| Goat | Anti-Mouse 568, IgG | Invitrogen | A-11031 | 1:200 |
| Goat | Anti-Rabbit 488, IgG | Invitrogen | A-11034 | 1:200 |
| Goat | Anti-Rabbit 568, IgG | Invitrogen | A-11036 | 1:500 |
| Goat | Anti-Mouse 488, IgG1 | Invitrogen | A-21121 | 1:500 |
| Goat | Anti-Mouse 568, IgG2b | Invitrogen | A-21144 | 1:200 |

Host, antibody name, company, category number and dilution are provided.
*Antibody provided by Manuel Koch. MyHCn: neonatal myosin heavy chain.

or NCAM were stitched into one seamless image using Fiji (ImageJ, v.1.51).

BrdU staining of the proliferating cells was not strong enough to analyse in a reliable manner so only mRNA data are provided for PRO. Differentiating cells, stained with desmin and myogenin, were imaged with an AxioScan.Z1 slide scanner (Carl Zeiss). A standardized region of interest (ROI), which covered approximately 90% of the coverslip, was defined (Fig. 3*A*). Damaged areas (due to handling of the coverslips) or large air bubbles were removed from the ROI before imaging. Images were captured using a plan-apochromat $10\times/0.45$ NA objective and a MultiBand filter cube (DAPI/FITC/TexasRed) using excitation wavelengths of 353, 493 and 577 nm (LED light source) and both coarse and fine focusing steps. Each channel was imaged separately and sequentially with an AxioCam MR R3 and a 10% overlap between images. Merged images were stitched using ZEN blue software (Carl Zeiss).

### Image analyses

The same person, blinded to group and leg, analysed all samples. The number of fibres included in each analysis is provided in Table 3.

**Myofibre size and type.** Myofibre cross-sectional area, type composition and type area percentage, were analysed on composite images (dystrophin/myosin/DAPI) using a semi-automated macro, run in Fiji, as described (Karlsen *et al.* 2019). Transversally cut myofibres were delineated and classified as type I, type II or hybrid based on median staining intensity. Hybrid fibres were detected in all three groups (0.9 [0–7.8]% in young, 0.5 [0–2.7]% in LLEX and 1.2 [0–5.6]% in SED) and were removed from the analysis. Myofibre type composition was also manually assessed on the same composite images by counting all visible type I, type II or hybrid myofibres using the ObjectJ plugin in Fiji. Myofibre-type composition obtained by manual counting and using the semi-automated macro were strongly correlated ($R^2 = 0.971$). Fibre-type area percentage was determined as a function of fibre-type percentage and fibre cross-sectional area (CSA).

**Satellite cells.** Satellite cells were manually quantified on composite images (laminin, Pax7, myosin I, DAPI) using the ObjectJ plugin in Fiji. Pax7$^+$ cells, also DAPI$^+$, were classified as satellite cells, and were allocated to type I or II fibres. If the 'parent' fibre could not be clearly identified, the respective satellite cell was marked separately, and later shared between fibre types. This occurred for 14 out of a

total of 4614 satellite cells counted. Satellite cell number was expressed relative to the number of fibres included in the analysis. Two samples were excluded from type II analysis due to a low number of fibres (SED control leg, $n = 14$ and LLEX exercised leg, $n = 15$).

**Denervated fibres.** The presence of MyHCn$^+$ and NCAM$^+$ fibres was manually assessed on composite images (dystrophin, NCAM/MyHCn, DAPI) using the ObjectJ plugin in Fiji. The CSA of all NCAM$^+$ fibres was measured and checked for co-expression of MyHCn and MyHC I. Then the CSA of all MyHCn$^+$ fibres was measured. Lastly, we removed all NCAM$^+$ or MyHCn$^+$ fibres that were not merosin$^+$ and desmin$^+$, or merosin$^+$ and phalloidin$^+$, as further confirmation that included cells were of myogenic origin. Fibres that had disappeared on a subsequent section or could not be convincingly located were marked separately as 'lost'.

**Cell culture.** The stitched images were separated into regions (2.26 × 1.80 mm, 3510 × 2790 pixels) equal to

$3 \times 3$ of the original image tiles of the slide scanner. As it was observed that cells were more densely located toward the centre of the coverslip, only regions within a central rectangular ROI on the coverslip were used. Automated thresholding of the DAPI channel was used to determine the approximate number of nuclei within each region and the region with a nuclei count closest to the median for that coverslip was selected for further analysis. As we had three technical replicates placed on separate plates, we analysed cells of the exercised and control leg that were cultured on the same plate. The next step included a manual correction of any mistakes made by the macro in delineating nuclei, e.g. fusing a single nucleus that had been split or separating several nuclei that were clumped together. Then the corrected nuclei were superimposed on the desmin channel, and nuclei that were located within myotubes with three nuclei or more were manually selected. Due to a small amount of bleed-through of desmin signal in the myogenin channel, the myogenin signal in each image was corrected by fitting the myogenin intensity *vs.* desmin intensity outside of nuclei (containing no true myogenin signal) and

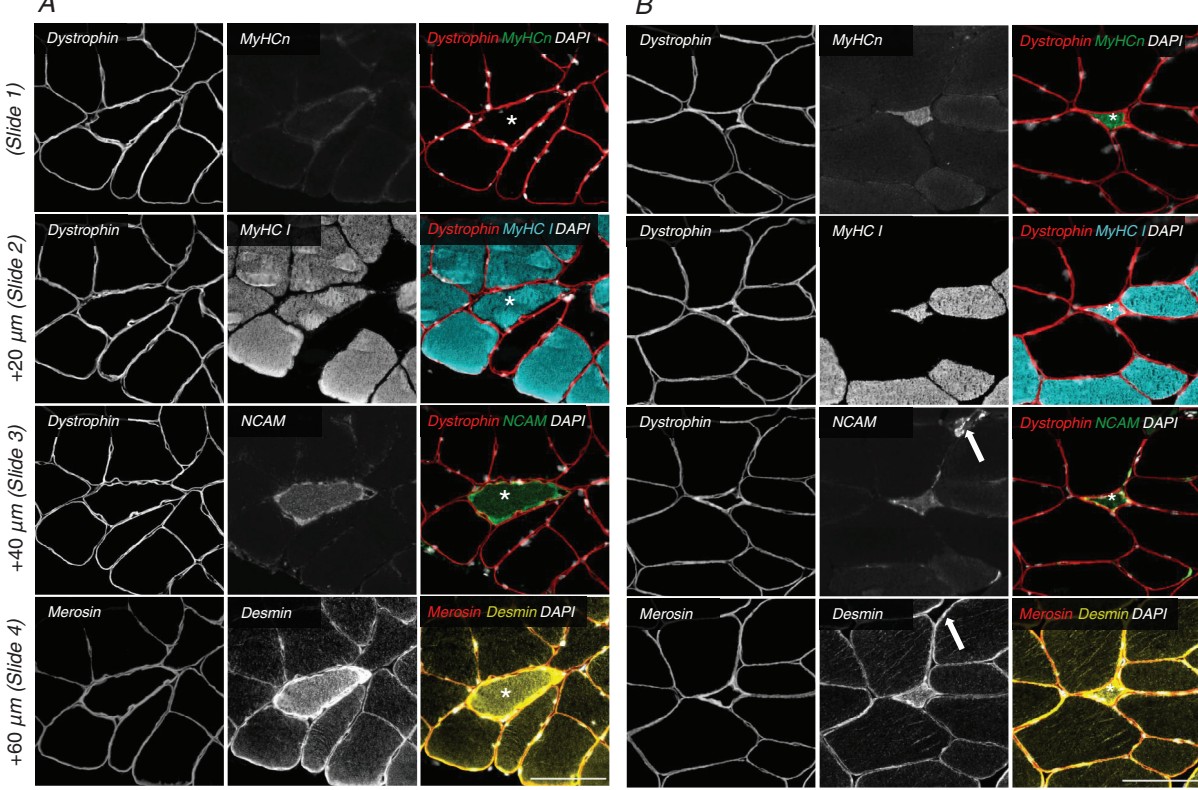

**Figure 2. Cross-sectional profiles of denervated fibres**
Split channel view of four serial sections from two vastus lateralis biopsies obtained from healthy elderly individuals. Sections have been stained with dystrophin + MyHCn (slide 1), dystrophin + MyHC I (slide 2), dystrophin + NCAM (slide 3) and merosin + desmin (slide 4). Notice in *A* the denervated fibre is positive for NCAM and more strongly positive for desmin than neighbouring fibres. Notice in *B* the NCAM signal in the upper right corner (arrows) which is not a myofibre (based on lack of staining for dystrophin, desmin and merosin). Asterisk in the merged image indicates the same denervated fibre on serial sections. Scalebars are 100 $\mu$m.

**Table 3. Number of fibres included in each image analysis**

| | Young | | LLEX | | SED | |
|---|---|---|---|---|---|---|
| **Myofibre CSA** | | | | | | |
| Type I | 227 ± 147 | 39–604 | 253 ± 118 | 79–485 | 248 ± 110 | 124–456 |
| Type II | 222 ± 71 | 72–314 | 157 ± 89 | 73–380 | 261 ± 127 | 69–539 |
| **Myofibre type composition** | | | | | | |
| Type I | 436 ± 272 | 303–1108 | 549 ± 253 | 170–1091 | 533 ± 219 | 303–1108 |
| Type II | 451 ± 173 | 169–721 | 324 ± 172 | 61–743 | 588 ± 275 | 233–1009 |
| **Satellite cells** | | | | | | |
| Type I, control leg | 245 ± 128 | 125–625 | 366 ± 185 | 129–837 | 338 ± 124 | 122–587 |
| Type I, exercised leg | 253 ± 131 | 110–621 | 391 ± 142 | 131–636 | 331 ± 173 | 97–587 |
| Type II, control leg | 276 ± 94 | 110–504 | 224 ± 94 | 80–387 | 400 ± 176 | 159–763 |
| Type II, exercised leg | 291 ± 108 | 145–521 | 270 ± 107 | 101–442 | 364 ± 189 | 76–712 |
| **Denervated fibres** | | | | | | |
| Control leg | 978 ± 369 | 423–1513 | 894 ± 340 | 361–1751 | 1099 ± 378 | 402–1688 |

*Values are given as average with standard deviations and ranges. Abbreviation: CSA, cross-sectional area.*

subtracting this fit from the intensity of the entire myogenin image. To improve homogeneity between samples with differing staining intensity, a contrast enhancement was performed on the desmin and myogenin channels. Data lists containing intensities in all channels for each nucleus were exported from Fiji and a custom MATLAB script (MATLAB R2019a, The MathWorks Inc.) was used for aggregating the data and determining desmin$^+$ and myogenin$^+$ cells by a threshold in the intensity of the respective channels within each nucleus. Area covered by myogenic cells (area of desmin$^+$ signal) was automatically measured. Fusion index was determined as the ratio of fused nuclei to desmin$^+$ nuclei, and differentiation index was determined as the ratio of myogenin$^+$ nuclei to

desmin$^+$ nuclei. Samples with a cell purity, determined as percentage desmin$^+$ cells, below 90% were removed from all data sets. Twelve of 92 samples (5/7 control/exercised leg and 1/7/4 young/SED/LLEX) were removed (Fig. 3*B*).

### Statistical analyses

Data are presented as means ± standard deviations or individual values with median unless stated otherwise in the figure legend. A significance level of $P < 0.05$ was chosen, with tendencies ($P < 0.1$) provided. Figures and tables were designed using Prism (v.8, GraphPad Software) and Excel 2016 (Microsoft), respectively. SigmaPlot (v. 13.0, Systat Software) was

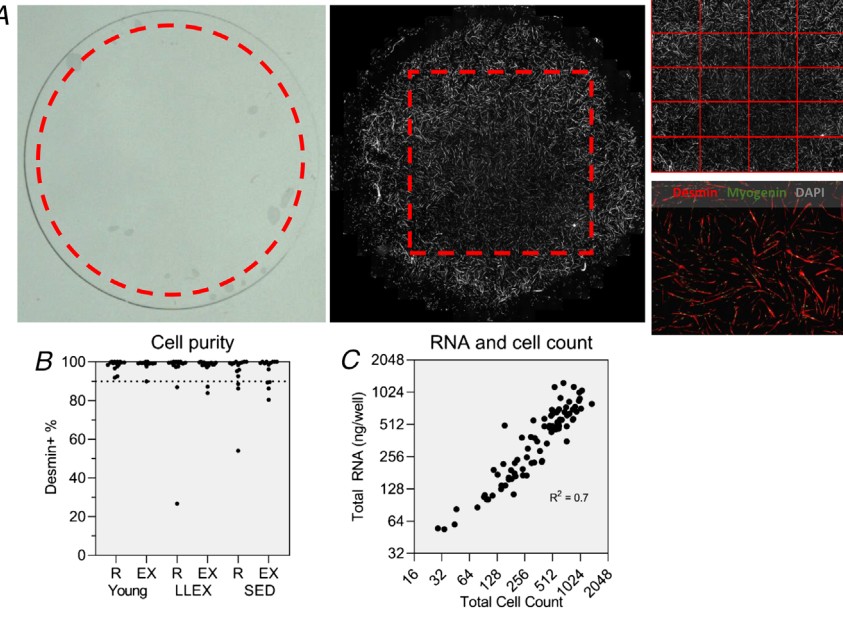

**Figure 3. Cell culture image analysis**
*A*, an example of an overview widefield microscope image, a stitched image covering entire coverslips with the rectangular ROI, a separation of regions each equal to 3 × 3 of the original image tiles of the slide scanner and a chosen region used for analyses. *B*, cell purity data from both control and exercise leg determined semi-automatically from DAPI and desmin stain. A threshold was set at 90% purity; samples below were excluded from all analyses. *C*, correlation between total cell count and total RNA for each sample (*n* = 90). [Colour figure can be viewed at wileyonlinelibrary.com]

**Table 4. Participant characteristics**

| | Young *vs*. old | LLEX *vs*. SED | Young n = 15 | | LLEX n = 16 | | SED n = 15 | |
|---|---|---|---|---|---|---|---|---|
| **Anthropometric** | | | | | | | | |
| Age (yr) | <0.0001 | 0.679 | 26 ± 5 | 20–36 | 73 ± 4 | 68–82 | 73 ± 4 | 68–82 |
| Height (cm) | 0.016 | 0.482 | 183 ± 7 | 169–193 | 176 ± 6 | 166–185 | 178 ± 8 | 161–195 |
| Weight (kg) | 0.365 | 0.124 | 82 ± 13 | 62–105 | 76 ± 9 | 63–94 | 82 ± 11 | 65–109 |
| BMI (kg/m$^2$) | 0.507 | 0.277 | 24 ± 3 | 20–30 | 24 ± 3 | 21–31 | 26 ± 3 | 22–32 |
| **Blood sample** | | | | | | | | |
| CRP (mg/L) | 0.052 | 0.525 | 1.3 ± 0.8 | 1.0–4.0 | 2.4 ± 2.1 | 1.0–9.0 | 3.0 ± 3.2 | 1.0–13.0 |
| HbA1c (mmol/L) | 0.001 | 0.989 | 5.5 ± 0.4 | 4.9–6.2 | 5.9 ± 0.3 | 5.5–6.8 | 5.9 ± 0.5 | 5.0–6.5 |
| **DEXA** | | | | | | | | |
| Leg LBM (kg) | 0.004 | 0.455 | 22.7 ± 2.9 | 18.3–27.4 | 20.6 ± 2.1 | 17.6–24.6 | 20.0 ± 2.4 | 16.7–25.8 |
| Total BMC (kg) | 0.792 | 0.881 | 3.1 ± 0.5 | 2.4–4.0 | 3.1 ± 0.3 | 2.6–3.8 | 3.1 ± 0.4 | 2.4–3.9 |
| Fat percentage | 0.356 | 0.006 | 24.7 ± 6.6 | 10.0–33.0 | 23.6 ± 6.4 | 12.2–33.6 | 29.9 ± 5.4 | 15.0–38.0 |
| Android fat mass (kg) | 0.151 | 0.016 | 1.7 ± 0.8 | 0.2–3.5 | 1.7 ± 1.0 | 0.5–3.5 | 2.6 ± 0.9 | 0.8–4.3 |
| **KinCom** | | | | | | | | |
| RFD30 ms (Nm/s) | <0.0001 | 0.702 | 2145 ± 921 | 742–3841 | 924 ± 572 | 269–1990 | 851 ± 432 | 346–2094 |
| RFD200 ms (Nm/s) | <0.0001 | 0.729 | 1232 ± 372 | 688–1836 | 784 ± 204 | 409–1062 | 759 ± 175 | 555–1272 |

Values are given as averages with standard deviations and ranges. Data were analysed using unpaired *t* tests. Specific *P* values are provided in italics in the table. Abbreviations: LBM, lean body mass; BMC, bone mineral content; BMI, body mass index; CRP, C-reactive protein; DEXA, dual energy x-ray absorptiometry; RFD, rate of force development.

used for statistical analyses. LLEX and SED were directly compared, and young was compared with the old groups combined. Within-group differences between rested and exercised leg was also compared. Cell culture data and NCAM/MyHCn analyses were not normally distributed, so non-parametric statistics were used (Mann–Whitney rank sum test and Wilcoxon's signed rank test). All remaining data appeared normally distributed (mRNA data after log-transformation), prompting the use of unpaired and paired *t* tests. Isometric strength tests performed before and after the exercise bout and log-transformed creatine kinase values were evaluated with one-way repeated measures ANOVA (Tukey *post hoc*) for each group. Data from the exercise bout were averaged into rounds and analysed using a two-way ANOVA (group x round) with the Holm-Sidak *post hoc* analysis.

## Results

### Participant characteristics and heavy resistance exercise

LLEX and SED did not differ in age, height, weight or BMI (*P* = 0.679, 0.482, 0.124 and 0.277, Table 4). Young had lower levels of C-reactive protein and HbA1c compared with old (*P* = 0.052 and 0.001, Table 4). Young were stronger and had a higher LBM than old (*P* < 0.0001 and *P* = 0.035), while LLEX had a lower fat percentage

than SED (*P* = 0.006, Fig. 4 and Table 4). Relative strength tended to be higher in LLEX compared with SED (*P* = 0.087, Fig. 4*B*).

Force produced, expressed relative to MVC, was lower in round 2 than round 1 in all groups, and LLEX produced force at a higher relative level across all sampled repetitions than both young and SED (Fig. 5*A*). There was a decline in MVC immediately following the exercise bout, and creatine kinase increased at day 2 in all groups (*P* < 0.0001, Fig. 5*B*, *C*).

### Myofibre size and denervation

LLEX had a larger proportion of type I fibres than SED (*P* = 0.033), while there was a tendency for young to have a lower proportion of type I fibres than old (*P* = 0.060, Fig. 6*B*). Fibres that were only weakly stained with MyHC I (hybrid fibres) were detected in low numbers in all three groups (0.9 [0–7.8]% in young, 0.5 [0–2.7]% in LLEX and 1.2 [0–5.6]% in SED). Given that hybrid fibres are common in aged muscle, and are composed of two or three distinctive MyHCs (Andersen *et al.* 1999), we removed these fibres from our analysis as our myosin I staining provided insufficient insight into the myosin composition. Fibre-type area followed a similar pattern to fibre-type distribution. Young had larger type II fibres than old (*P* < 0.0001), while their type I fibres tended to be larger (*P* = 0.072, Fig. 6*C*). Both old groups had smaller type II

fibres compared with their own type I fibres (LLEX, $P = 0.003$, SED, $P = 0.015$). Myofibre morphology is illustrated in histograms, where the type II fibres of the old participants have shifted leftwards (Fig. 6*A*).

The percentage of NCAM$^+$ and MyHCn$^+$ fibres was larger in old than young ($P = 0.003$ and $0.034$), while no difference was observed between LLEX and SED ($P = 0.984$ and $0.352$, Fig. 7*A,B*). NCAM$^+$ fibres were classified as pure type I or II myofibres, or hybrids, with almost even numbers of types I and II. Furthermore, between 10 and 30% of NCAM$^+$ fibres co-expressed MyHCn (Fig. 7*D*). A large part of the NCAM$^+$ and MyHCn$^+$ fibres were <500 $\mu$m$^2$ (Fig. 7*C*). We observed an area in a sample that was reminiscent of MTJ, similar to what we have previously described (Soendenbroe *et al.* 2020). Control stainings with COL22 revealed that 14 out of 37 NCAM$^+$ fibres from that biopsy were related to the MTJ and were removed. A median (range) of $4.5 \pm 4.6$ and $1.8$ $(0–9) \pm 2.1$ fibres initially included in the NCAM and MyHCn counts, respectively, were removed following assessment for merosin, desmin and phalloidin (26 and 28% reduction in NCAM$^+$ and MyHCn$^+$ fibres, respectively). It was predominantly the very small myofibres that could not be detected on serial sections.

### Satellite cells and cell culture

In the control leg, LLEX had a greater number of type II myofibre-associated satellite cells than SED ($P = 0.016$), while no difference was observed for type I fibres ($P = 0.609$, Fig. 8*A*). Young had more satellite cells associated with both type I and II fibres, compared with old ($P = 0.035$ and $P < 0.0001$, Fig. 8*A*). LLEX and SED had fewer type II-associated satellite cells than type I ($P < 0.0001$ and $P = 0.006$, Fig. 8*A*). No difference in differentiation index was observed ($P = 0.695$), while a tendency for a higher fusion index in young compared with old was found ($P = 0.091$, Fig. 9*A*). Young had a higher cell count than old ($P = 0.002$), and a tendency for an increased desmin area in young compared with old was also observed ($P = 0.081$, Fig. 9*A*) We observed no effect of acute exercise on satellite cell number, differentiation index or fusion index, cell count or desmin area ($P$ values ranged from $0.094$ to $0.922$, Figs 8*B* and 9*B*).

### Gene expression

At the tissue level AChR $\delta$, $\alpha1$ (tendency) MuSK and MyHCn mRNA were lower in young compared with old ($P = 0.047$, $0.086$, $0.014$ and $P < 0.0001$, Fig. 10*A*). AChR $\beta1$ and $\gamma$ were higher in LLEX compared with SED ($P = 0.022$ and $0.026$, Fig. 10*A*). MyHCe gene expression was upregulated in the exercised leg of LLEX ($P = 0.035$), and AChR $\alpha1$ and MuSK tended to be expressed higher in the exercised leg of LLEX and young, respectively ($P = 0.098$ and $0.074$, Fig. 10*A*).

In proliferating myoblasts COL1a1 and p16 were lower, and myogenin, MyHCn and MyHCe higher in young compared with old ($P = 0.047$, $0.016$, $0.001$, $0.018$ and $0.013$, Fig. 10*B*). Myogenin tended to be lower in LLEX than SED ($P = 0.078$, Fig. 10*B*). In differentiating myotubes, p16 was lower, and AChR $\gamma$, MyHCn and MyHCe were higher in young compared with old ($P = 0.0001$, $0.002$, $0.024$ and $0.035$, Fig. 10*C*). No differences between LLEX and SED were observed (Fig. 10*C*). Similarly, no effects of acute exercise were in proliferating or differentiating cells for either group (Fig. 10*B, C*).

### Discussion

Skeletal muscle of lifelong recreationally active elderly individuals retains a higher number of type II fibre-associated satellite cells, possesses a beneficial innervation status when assessed by RT-qPCR, and performs substantially better during acute resistance exercise, compared with sedentary individuals. These findings indicate that lifelong recreational activity can partially offset the emergence of classic phenotypic traits associated with the aged muscle.

### *In vivo* measure of muscle function

The acute exercise bout caused a pronounced decline in force output both within sets and between sets

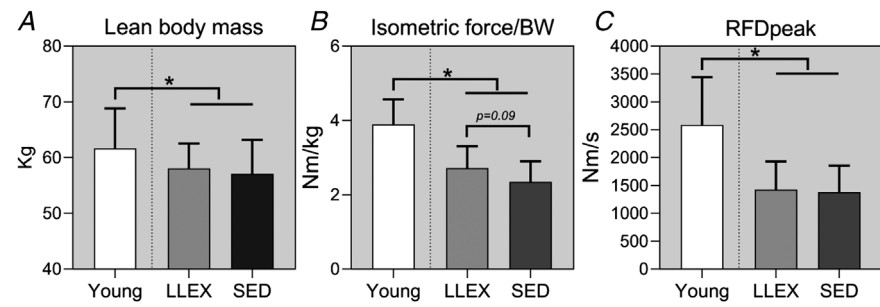

**Figure 4. Muscle mass and strength**
*A*, lean body mass, *B*, relative strength and *C*, rate of force development is provided for each group as averages with standard deviations. $n = 15$ (young), 16 (LLEX) and 15/14 (SED). Data were analysed using unpaired *t* tests. *$P < 0.05$ *vs.* young. Tendencies are written.

for all groups. Strikingly, LLEX outperformed both SED and the young group, confirming their status as exercise-habituated individuals. Despite this, no differences were observed in LBM or MVC between the old groups, indicating that these standard assessments may not allow subtle differences to be detected. Studies investigating the impact of a recreationally active lifestyle on LBM and MVC are inconclusive. When heavy resistance exercise is performed, or the participants are at the pinnacle of sporting performance within their age group in a strength or explosive type of event, then both muscle mass and function will have increased

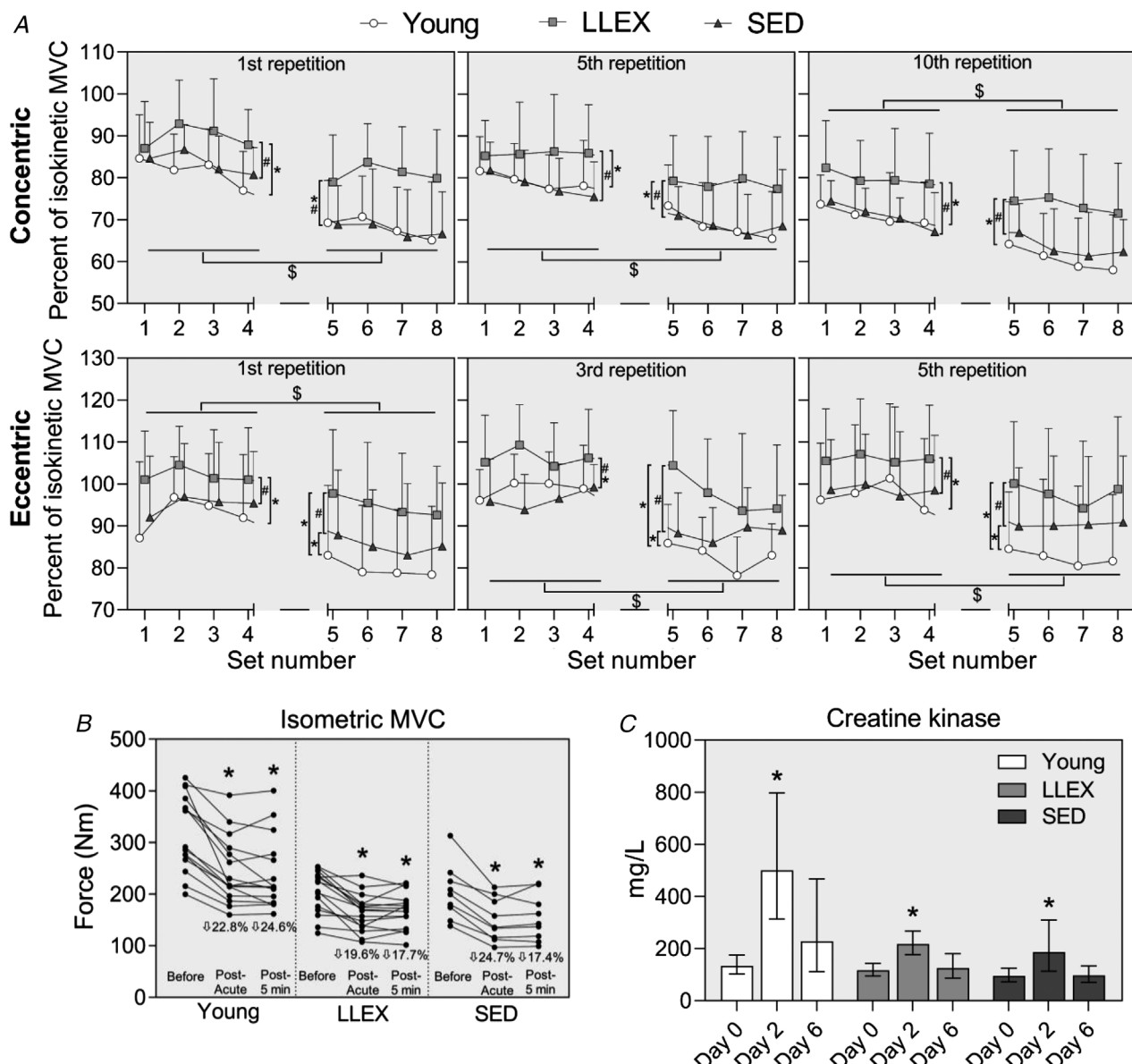

**Figure 5.  Acute bout of heavy resistance exercise**
*A*, the first, fifth and 10th concentric repetitions and the first, third and fifth eccentric repetition from each set was sampled during the exercise bout. Maximum torque values are expressed relative to concentric isokinetic MVCs and are shown as averages with standard deviations. *n* = 15 (young), 16 (LLEX) and 15 (SED). Set 1−4 (round 1) and 5−8 (round 2) average values were statistically evaluated using two-way ANOVA (group *x* round) with the Holm-Sidak *post hoc* analysis. # *P* < 0.05 *vs.* SED, **P* < 0.05 *vs.* young, $ *P* < 0.05 round two *vs.* round 1. *B*, isometric MVC, shown as individual values, before, immediately after the exercise bout and following a 5 min rest period. *n* = 15 (young), 16 (LLEX) and nine (SED). Data were analysed using one-way RM ANOVA with Tukey's *post hoc* analysis **P* < 0.05 *vs.* before. *C*, creatine kinase was measured on days 0, 2 and 6 and is shown as geometric mean with 95% CI. *n* = 15 (young), 15 (LLEX) and 14 (SED). Data were analysed using one-way RM ANOVA with the Tukey *post hoc* analysis. **P* < 0.05 *vs.* before/day 0. Abbreviations: MVC, maximal voluntary contraction.

accordingly (Klitgaard *et al.* 1990; Ojanen *et al.* 2007; Unhjem *et al.* 2016; Sonjak *et al.* 2019). On the other hand, several studies investigating recreationally active individuals have seen limited effects on LBM and MVC (Klitgaard *et al.* 1990; Lanza *et al.* 2008; Unhjem *et al.* 2016; St-Jean-Pelletier *et al.* 2017), suggesting that these measures are unable to discriminate between recreationally active and sedentary individuals of similar age. Accordingly, it is only under challenged conditions that functional differences become apparent between recreationally active and inactive elderly individuals. In support of this notion, a recent study found no correlation between daily steps and *in vivo* measurements of muscle function, except during challenged conditions in elderly men and women (Varesco *et al.* 2022).

## Satellite cell quantity and function

One of the main novel findings of the present study is the difference in type II myofibre-associated satellite cells between the physically active and inactive elderly men. Satellite cells are the sole source of new myonuclei and

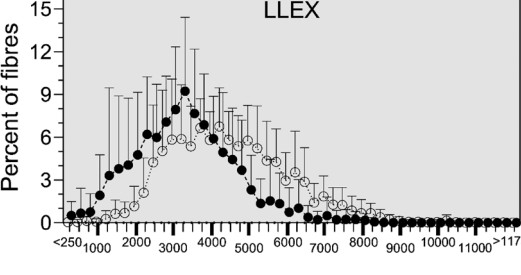

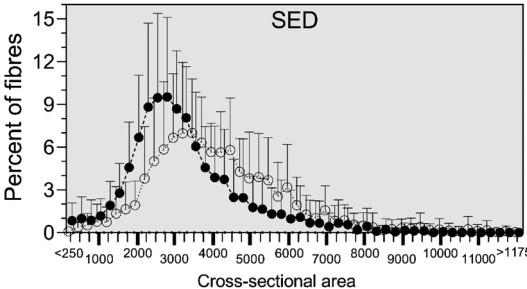

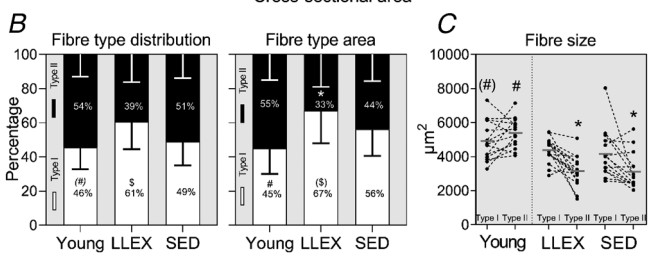

**Figure 6. Muscle morphology**
*A*, types I (open circle, dotted line) and II (filled circle, stippled line) fibre-size distribution shown as averages with standard deviations. *B*, fibre-type distribution and fibre-type area shown as averages with standard deviations. *C*, fibre size of types I and II shown as connected individual values and averages. *n* = 15 (young), 16 (LLEX) and 15 (SED). *Significantly different from type I within group, # significantly different from old, (#) tendency for a difference from old, $ significantly different from SED, ($) tendency for a difference from SED.

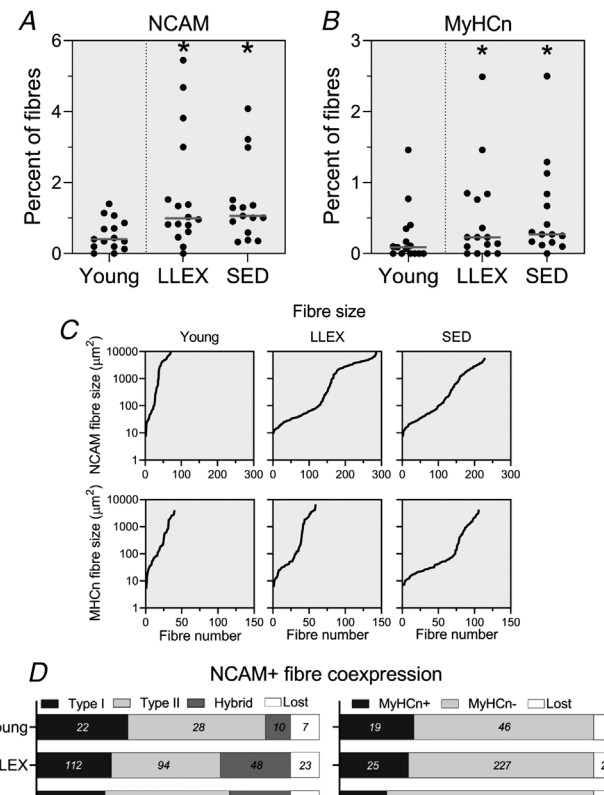

**Figure 7. Muscle innervation**
*A*, percentage of fibres expressing NCAM shown as individual values and median. *n* = 15 (young), 16 (LLEX) and 15 (SED). Data were analysed using a Mann–Whitney rank sum test. *Significantly different from young. *B*, percentage of fibres expressing MyHCn shown as individual values and median. *C*, fibre size given in $\mu m^2$ of all NCAM$^+$ and MyHCn$^+$ positive fibres for each group (y-axis is logarithmic). *D*, coexpression of NCAM fibres. Left shows the percentage of NCAM$^+$ fibres that are types I, II, hybrid or not found (lost). Right shows the percentage of NCAM$^+$ fibres that are MyHCn$^+$, MyHCn$^-$ or not found (lost). Number in each bar is the absolute number of fibres in that category.

are important not only for long-term muscle growth by facilitating accretion of myonuclei (Kadi *et al.* 2004; Fry *et al.* 2014) but also for inter-cell communication (Murach et al. 2021*b*) and NMJ maintenance (Liu *et al.* 2017). Satellite cell quantity is reduced with ageing (Karlsen *et al.* 2020), disease (Verdijk *et al.* 2012) and inactivity (Arentson-Lantz *et al.* 2016) and increased with acute (Heisterberg *et al.* 2018) and long-term (Kadi *et al.* 2004) exercise and during muscle regeneration (Karlsen *et al.* 2020). Type II myofibre-associated satellite cells are more severely affected by ageing than type I (Verdijk *et al.* 2014; Karlsen *et al.* 2019, 2020), but this decline could also be attributed to a reduced type II myofibre activation with ageing. The larger type II fibre satellite cell pool in LLEX thus provides a larger capacity to mount a myogenic response in the event of injury or denervation (Shefer *et al.* 2006), while simultaneously secreting signals taken up by muscle fibres and single-nucleated cells in or around the satellite cell niche (Murach et al. 2021*b*). Surprisingly, the exercise bout did not lead to an increase in satellite cell content, which might be related to the exclusive use of slow contractions, timing of biopsy sampling or insufficient stimulus (Hyldahl & Hubal, 2014; Snijders *et al.* 2015). To explore the function of the satellite cells, we performed cell culture studies and compared the capacity of satellite cells to differentiate and fuse, in addition to measuring mRNA levels of genes related to myogenesis and muscle innervation. Importantly, cultivated myogenic satellite cells have been shown to retain intrinsic capabilities reminiscent of their former *in vivo* environment (Teng & Huang, 2019). Contrary to our hypothesis, the two primary measures of cell function, differentiation and fusion index, were similar in LLEX and SED, while only a tendency for an age-related difference for fusion index

was observed, which might be explained by a higher cell number. Satellite cell proliferation could not be assessed due to problems relating to the staining protocol, so we cannot rule out potential differences between groups in myoblast proliferation. The literature on whether ageing affects satellite cell function in culture is mixed, as some studies indicate phenotypic differences (Bechshøft *et al.* 2019; Balan *et al.* 2020) while others do not (Alsharidah *et al.* 2013; Chaillou *et al.* 2020). For example, we recently showed that the fusion capabilities were reduced in old compared with young subjects (Bechshøft *et al.* 2019), while Chaillou *et al.* 2020 found no difference in fusion index or myotube diameter between young and old (Chaillou *et al.* 2020). The cause of these discrepancies between studies is unclear but may at least partly be due to differences in the employed cell culture models (cell lines or primary cells) or the immunofluorescence and image analyses. As such, a strength of the present study is that entire coverslips were imaged, which allowed the analysis of areas with the most representative cell presence, and that technical replicates were used. In line with our earlier studies, several age-related differences in gene expression of proliferating and differentiating satellite cells were observed (AChR $\gamma$ subunit, myogenin, COL1A1, MyHCn, MyHCe and p16) (Bechshøft *et al.* 2019; Soendenbroe *et al.* 2020). No significant differences were observed between LLEX and SED. Overall, the satellite cell data are supportive of age-related differences in both satellite cell quantity *in vivo*, measured by immunofluorescence microscopy, and function *in vitro*, as evidenced by differences in the gene expression of several genes related to myogenesis and muscle innervation. However, neither differentiation nor fusion index, the primary measures of cell function, were affected by age, although the

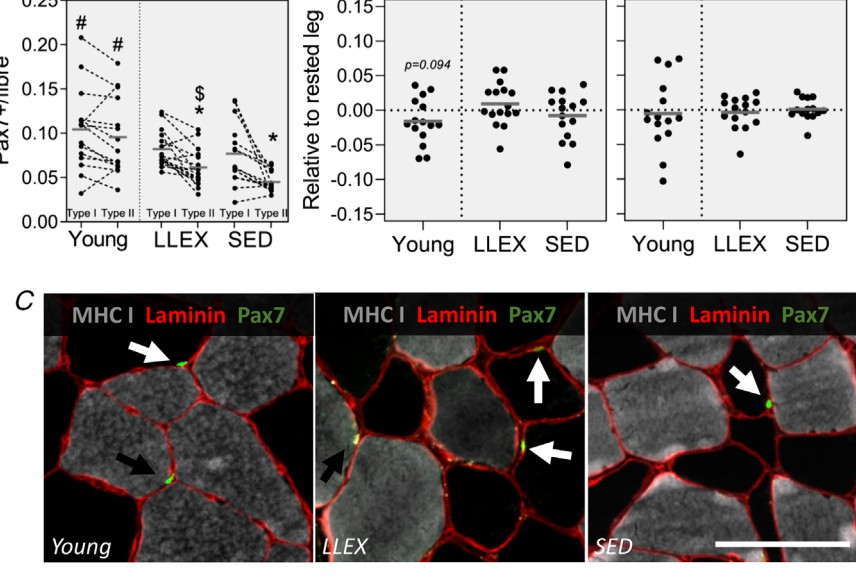

**Figure 8. Satellite cell quantity**
*A*, satellite cells per fibre in control leg given as individual values for both type I and II fibres (connected by dashed line) with average value (horizontal line). Data were analysed using unpaired *t* tests. *B*, exercise response in type I and II satellite cells shown as individual values and averages (horizontal line). Data were analysed using paired *t* tests. *n* = 15, 16 and 15 for young, LLEX and SED. *Significantly different from type I within group, # significantly different from old, $ significantly different from SED. Tendencies are written. *C*, example of type I (black arrow) and type II (white arrow) myofibre-associated satellite cells in young (left), LLEX (middle) and SED (right). Scalebar is 100 $\mu$m. [Colour figure can be viewed at wileyonlinelibrary.com]

influence on proliferation remains to be determined. Life-long recreational exercise affected satellite cell numbers positively, while no change in satellite cell function was observed. Next, we wanted to know if these differences amounted to differences in muscle innervation status and myofibre morphology.

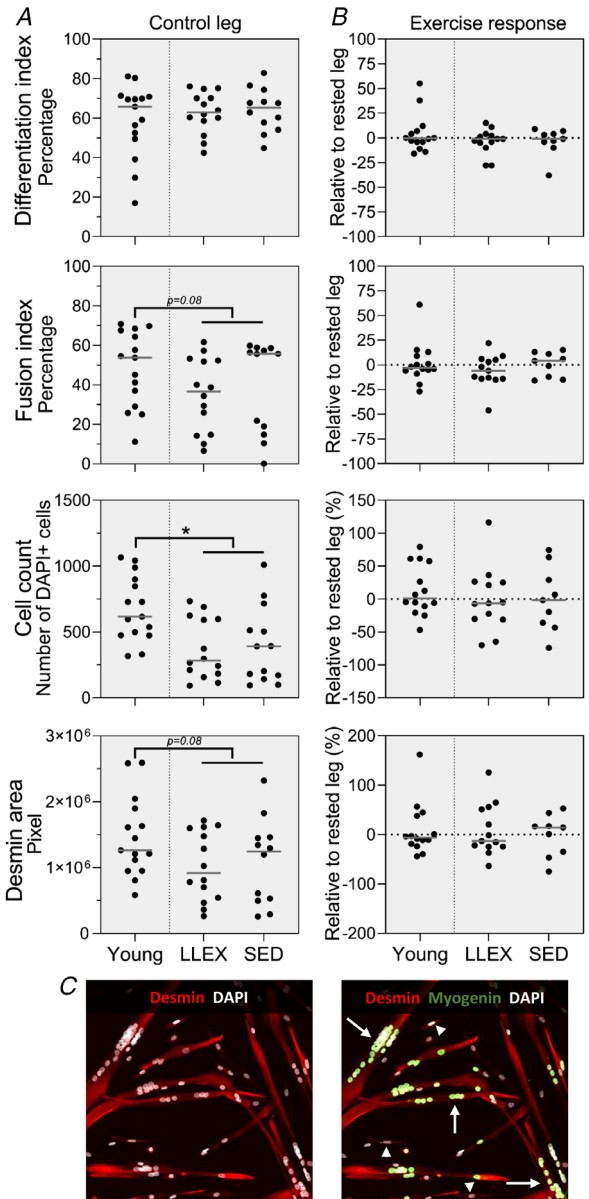

**Figure 9. Satellite cell function**

*A*, differentiation index, fusion index, cell count and desmin area of human myogenic cells cultured for 7 days shown as individual values with median line (*n* = 15, 14 and 12 for young, LLEX and SED). Data were analysed using Mann–Whitney's rank sum test. *B*, exercise response in differentiation index, fusion index, cell count and desmin area shown as individual values with median line (*n* = 14, 13 and 9 for young, LLEX and SED. Data were analysed using Wilcoxon's signed rank test. *Significantly different from old. Tendencies are written. *C*, representative example of cell culture; arrows and arrowheads point to fused and non-fused nuclei, respectively. [Colour figure can be viewed at wileyonlinelibrary.com]

## Muscle innervation status

Innervation status was assessed by immunofluorescence microscopy and RT-qPCR analyses. Both methods were used as they might represent myofibres at different stages of denervation or differ in how they are regulated. NCAM and MyHCn were used as IHC markers for denervated myofibre, as we (Soendenbroe *et al.* 2019, 2020) and others (Mosole *et al.* 2014; Sonjak *et al.* 2019; Daou *et al.* 2020; Burke *et al.* 2021; Monti *et al.* 2021) have previously done. It should be noted that NCAM and MyHCn are also associated with other physiological processes and structures within muscle, which can challenge the interpretation. NCAM is found at the NMJ and MTJ (Moore & Walsh, 1985; Jakobsen *et al.* 2018), during muscle regeneration (Irintchev *et al.* 1994; Mackey & Kjaer, 2017) and in neuromuscular disease (Walsh & Moore, 1985). MyHCn is found during muscle regeneration (Sartore *et al.* 1982; Mackey & Kjaer, 2017), in neuromuscular disease (Fitzsimons & Hoh, 1981) and in intrafusal fibres (Walro & Kucera, 1999). However, in healthy vastus lateralis muscle tissue, MTJ and NMJ structures are easily recognized, intrafusal fibres are rare, and muscle regeneration is unlikely to be present. Furthermore, experimentally induced muscle denervation leads to a large upregulation in the expression of NCAM and MyHCn (Covault & Sanes, 1985; Schiaffino *et al.* 1988), together making muscle fibre denervation the most likely explanation for the observation of NCAM[+] and MyHCn[+] fibres in our study. In accordance with our previous findings (Soendenbroe *et al.* 2020), old subjects had a higher number of NCAM[+] and MyHCn[+] fibres than young. However, in contrast to our hypothesis, we did not see indications of favourable innervation status in the LLEX group using our immunofluorescent approach. Importantly, several novel findings relating to exercise status were observed in the gene expression data. LLEX had significantly higher mRNA levels of both AChR $\beta 1$ and $\gamma$ subunits compared with SED, and young had lower AChR $\delta$ and $\alpha 1$ (tendency) compared with old. AChR gene expression has been reported to be affected by disease (Kapchinsky *et al.* 2018; Kelly *et al.* 2018), injury (Gigliotti *et al.* 2015; Karlsen *et al.* 2020), ageing (Spendiff *et al.* 2016; Soendenbroe *et al.* 2020), inactivity (Monti *et al.* 2021) and acute exercise (Soendenbroe *et al.* 2020). Given the remarkable similarity of the AChR gene expression profile between LLEX and the young group, it could be speculated that the young group might have been habitually more active than SED, which would push them in the direction of LLEX. Activity levels are well known to change with ageing (Hallal *et al.* 2012). As previously mentioned, very few human studies examine human AChRs, and this is the first study to report data for all muscle-specific AChR subunits in lifelong recreationally active elderly men. Overall, it appears that the analysis of

AChR gene expression is more sensitive than the currently available immunofluorescent markers of denervation. But the use of immunofluorescent markers in the present study has added important details on the morphology of the denervated fibres. Most denervated fibres are very small, often with a CSA of less than a tenth of the mean normal fibre size of the elderly groups. Furthermore,

the rigorous assessment required the presence of several myogenic markers such as a dystrophin (sarcolemma), merosin (basal lamina), MyHC and desmin, as well as containing general cell actin. Approximately similar proportions of the denervated fibres in LLEX and SED are type I, II and hybrid fibres. Interestingly, the overlap between the used markers (NCAM and MyHCn)

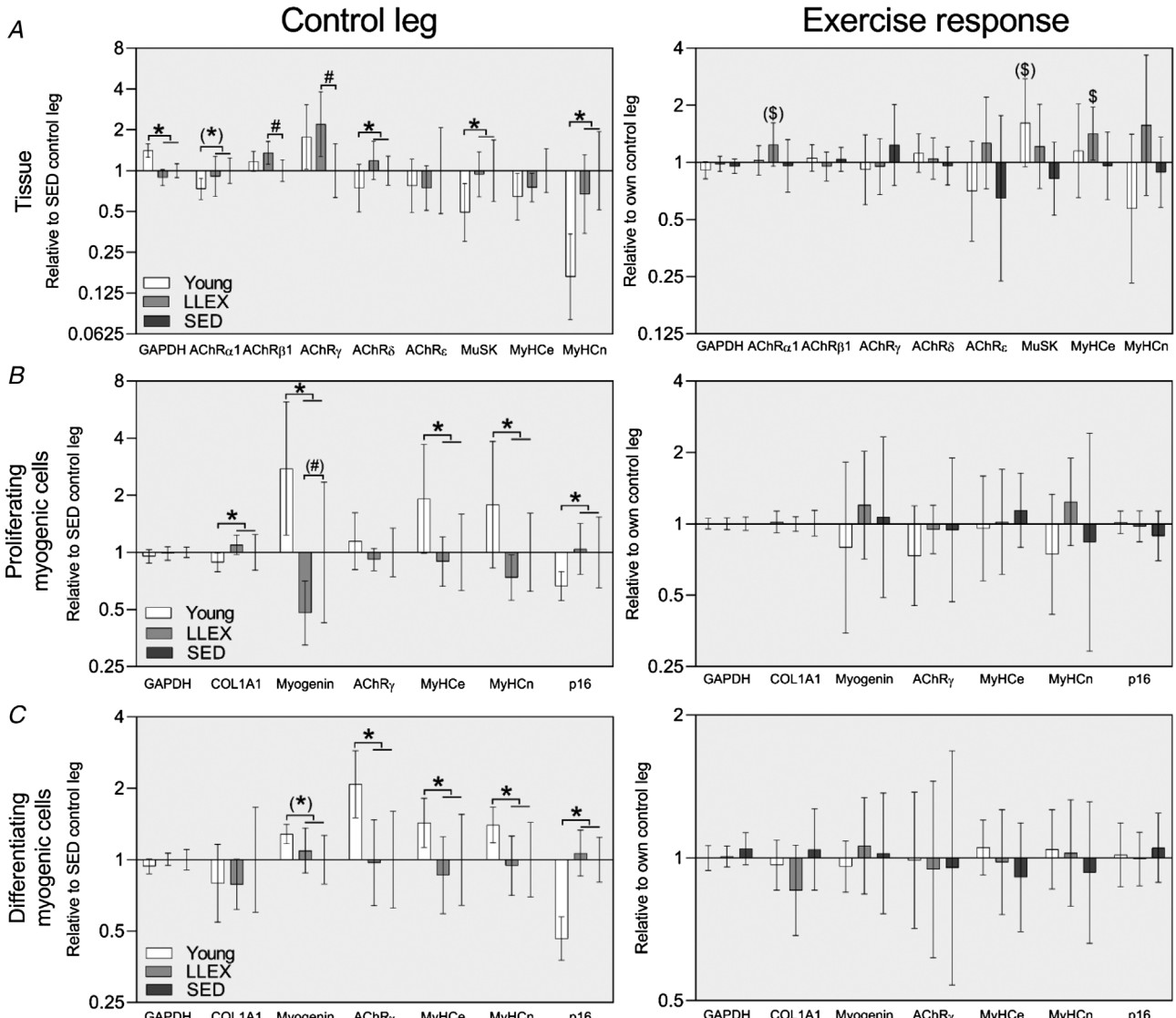

**Figure 10. Gene expression in biopsies and cells**
*A*, gene expression in muscle biopsies (*n* = 15, 16 and 15 for young, LLEX and SED). *B*, proliferating myoblasts (*n* = 14/13, 15/14 and 12/8 for control/exercise leg of young, LLEX and SED). *C*, differentiating myotubes (*n* = 15/14, 14/13 and 14/11 for control/exercise leg of young, LLEX and SED). Control (left) and exercised (right) leg. mRNA data were normalized to RPLP0 and are shown as geometric means with 95% confidence intervals. Control leg is shown relative to SED control leg and exercise response is shown relative to own control leg. Baseline differences were analysed using unpaired *t* tests, and exercise responses were analysed using paired *t* tests. *$P < 0.05$ young *vs.* old. (*) $P < 0.1$ young *vs.* old. # $P < 0.05$ LLEX *vs.* SED. (#) $P < 0.1$ LLEX *vs.* SED. $ $P < 0.05$ exercised *vs.* control leg. ($) $P < 0.1$ exercised *vs.* control leg.

was limited, which might be due to temporal variation in protein expression of denervated fibres, or that subgroups of denervated fibres exist. Also, MyHCn positive fibres have a segmented staining profile, which, due to the cross-sectional approach, could also explain at least a portion of the discrepancy (Schiaffino *et al.* 1988; Soendenbroe *et al.* 2019, 2021).

Lastly, myofibre morphology was comprehensively studied as it ties closely with both innervation status and satellite cell numbers. We found, as expected, that the young group had larger type II and type I (tendency) fibres than the old groups combined, which has been shown before (Klitgaard *et al.* 1990; Zampieri *et al.* 2015; St-Jean-Pelletier *et al.* 2017; Karlsen *et al.* 2019; Sonjak *et al.* 2019). In contrast to our hypothesis, however, no difference in fibre size was observed between LLEX and SED. The reason for the lack of difference in average fibre size is unclear, but it is possible that the activities performed by the individuals of the LLEX group did not possess a large enough hypertrophic stimulus for the fibres to increase in size. In general, heavy loading has been shown to be crucial for type II myofibre hypertrophy (Klitgaard *et al.* 1990), and endurance exercise has a limited effect on type II fibre CSA (McKendry *et al.* 2020). Ten of the subjects in the present study reported performing resistance exercise, although some of these only did it once a week, some only during the off season of their primary activity and some with light loads. Only three subjects reported resistance exercise as their primary activity. Since muscle loading, volume and training frequency are all major determinants of hypertrophy, it is likely that the activities performed have not forced an adaptation in myofibre size. It is also noteworthy that while the amount and type of activity performed by LLEX did not appear to preserve type II myofibre size, it was associated with a preservation of the number of type II myofibre-associated satellite cells, suggesting that these two entities are not tightly regulated in healthy elderly muscle. The study of master athletes remains a suitable model to study ageing disentangled from physical inactivity (Harridge & Lazarus, 2017). However, the number of individuals performing exercise at a level where they can be considered master athletes is low (Hallal *et al.* 2012), which coincidentally makes the study of recreationally active individuals more relevant. All individuals in the present study were independent and well-functioning, meaning that a decline in muscle function would be expected in the years ahead. It has been shown that the muscle of very old individuals remains amenable to improvement (Kryger & Andersen, 2007), indicating that although few differences between the groups were observed, the recreationally active individuals might be on a different trajectory, which could benefit them later in life when phenotypic traits of the aged muscle are more pronounced.

## Conclusion

Recreational physical activity preserves type II myofibre-associated satellite cells during ageing, and leads to a more beneficial muscle innervation status. These data strongly suggest that detrimental effects of ageing can be partially offset by lifelong self-organized recreational exercise. Furthermore, this is the first attempt in humans to investigate satellite cells and myofibre denervation in parallel, and how they are each influenced by exercise. The study is limited by the lack of objective measures of levels of physical activity and the inclusion of only male participants. In our earlier study on young and elderly females, similar findings on myofibre denervation were reported (Soendenbroe *et al.* 2020). Clearly, studies of lifelong exercise in females are needed. The translational perspective of the present study is heightened due to the focus on recreationally active individuals rather than master athletes, as the former constitute a far larger part of the general population aged 60 and above.

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

## Additional information

### Data availability statement

mRNA data from tissue and cells can be found in online Supplementary material. Additional original data can be provided, in an anonymized manner, to interested parties. Contact C.S. and A.L.M. and describe the specific data that are needed and the intended use of the data.

### Competing interests

None.

### Author contributions

A.L.M. and C.S. contributed to the first hypothesis generation and A.L.M. provided resources. C.S., P.S., M.K., J.L.A. and A.L.M. contributed to the conceptual design and A.L.M. and J.L.A. supervised this work. C.S., M.T., R.B.S., P.S., J.L.A. and A.L.M. developed the methodology. C.S., C.M. and M.T. performed the experiments and C.S., C.L.D., R.B.S. and P.S. performed data analysis and visualization. C.S., R.B.S., P.S., M.K., J.L.A. and A.L.M. performed the analysis and data interpretation and C.S. and A.L.M. wrote the manuscript. All authors edited and reviewed the manuscript.

### Funding

We gratefully acknowledge grants from The Lundbeck Foundation (R344-2020-254), the Nordea Foundation (Centre for Healthy Aging), the Danish Agency for Culture (FPK.2018-0036), the AP Møller Foundation for the Advancement of Medical Science and Copenhagen University Hospital – Bispebjerg and Frederiksberg.

### Acknowledgements

We acknowledge the Core Facility for Integrated Microscopy, Faculty of Health and Medical Sciences, University of Copenhagen, where the AxioScan.Z1 slide scanner images were obtained. The Department of Clinical Biochemistry, Bispebjerg Frederiksberg Hospital, University of Copenhagen, Copenhagen, Denmark, is acknowledged for analysing the blood samples.

The monoclonal antibodies A4.951 (myosin heavy chain, human slow fibres), BA-D5 (myosin heavy chain, human slow fibres), PAX7 and F5D (myogenin), developed by Blau H.M., Schiaffino S., Kawakami A. and Wright W.E., respectively, were obtained from the Developmental Studies Hybridoma Bank, created by the NICHD of the NIH, and maintained at The University of Iowa, Department of Biology, Iowa City, IA 52242. The collagen 22 antibody was kindly provided by Manuel Koch.

The authors thank Anja Jokipii-Utzon and Ann-Christina Ronnié Reimann for excellent technical assistance with preparation of the muscle biopsies and the mRNA analysis.

### Keywords

acetylcholine receptor, denervation, human skeletal muscle, lifelong exercise, sarcopenia, satellite cells

## Supporting information

Additional supporting information can be found online in the Supporting Information section at the end of the HTML view of the article. Supporting information files available:

**Peer Review History**
**Statistical Summary Document**
**Data S1**
**Data S2**

