## [Peer Review History · The Journal of Physiology]

Preserved stem cell content and innervation profile of elderly human skeletal muscle with lifelong recreational exercise

Casper Soendenbroe, Christopher Lund Dahl, Christopher Meulengracht, Michal Tamáš, Rene B. Svensson, Peter Schjerling, Michael Kjaer, Jesper Løvind Andersen, and Abigail Louise Mackey

DOI: 10.1113/JP282677

Corresponding author(s): Casper Soendenbroe (casper.soendenbroe@regionh.dk)

The following individual(s) involved in review of this submission have agreed to reveal their identity: Davis Englund (Referee #1); Gianni Parise (Referee #2)

Review Timeline:

Submission Date:	02-Dec-2021
Editorial Decision:	04-Jan-2022
Revision Received:	01-Feb-2022
Accepted:	14-Feb-2022

Senior Editor: Richard Carson

Reviewing Editor: Kevin Murach

Transaction Report:

Dear Dr Soendenbroe,

Re: JP-RP-2021-282677 "Superior stem cell content and innervation profile of elderly human skeletal muscle with lifelong recreational exercise" by Casper Soendenbroe, Christopher Lund Dahl, Christopher Meulengracht, Michal Tamáš, Rene B. Svensson, Peter Schjerling, Michael Kjaer, Jesper Løvind Andersen, and Abigail Louise Mackey

Thank you for submitting your manuscript to The Journal of Physiology. It has been assessed by a Reviewing Editor and by 2 expert referees and I am pleased to tell you that it is considered to be acceptable for publication following satisfactory revision.

The reports are copied at the end of this email. Please address all of the points and incorporate all requested revisions, or explain in your Response to Referees why a change has not been made.

NEW POLICY: In order to improve the transparency of its peer review process The Journal of Physiology publishes online as supporting information the peer review history of all articles accepted for publication. Readers will have access to decision letters, including all Editors' comments and referee reports, for each version of the manuscript and any author responses to peer review comments. Referees can decide whether or not they wish to be named on the peer review history document.

I hope you will find the comments helpful and have no difficulty returning revisions within 4 weeks.

If you need to check to make sure that your Methods section conforms to the principles of UK regulations, you may wish to refer to Grundy (2015):
Grundy (2015) J. Physiol. 2015 Jun 15;593(12):2547-9 <https://doi.org/10.1113/JP270818>

Your revised manuscript should be submitted online using the links in Author Tasks Link Not Available. This link is to the Corresponding Author's own account, if this will cause any problems when submitting the revised version please contact us.

The image files from the previous version are retained on the system. Please ensure you replace or remove any files that have been revised.

REVISION CHECKLIST:

- Summary data must be reported as mean {plus minus} SD or 95% confidence interval
- All table and figure legends with summary data must include the statistical test used in the table/figure and sample size
- Figures with summary data bars must include individual data points, or box whisker plots when $n > 30$.
- Article file, including any tables and figure legends, must be in an editable format (eg Word)
- Upload each figure as a separate high quality file
- Upload a full Response to Referees, including a response to any Senior and Reviewing Editor Comments;
- Upload a copy of the manuscript with the changes highlighted.

- A potential 'Cover Art' file for consideration as the Issue's cover image;
- Appropriate Supporting Information (Video, audio or data set https://jp.msubmit.net/cgi-bin/main.plex?form_type=display_requirements#supp).

To create your 'Response to Referees' copy all the reports, including any comments from the Senior and Reviewing Editors, into a Word, or similar, file and respond to each point in colour or CAPITALS and upload this when you submit your revision.

I look forward to receiving your revised submission.

If you have any queries please reply to this email and the Peer Review Coordinator will be pleased to advise.

If revision is not possible, or if you cannot respond to the requests for change, contact us by return email as soon as possible, giving reasons for the difficulties. Withdrawal of the manuscript may be necessary in these circumstances, and instruction will be given on how to proceed. Please note that a paper must be withdrawn before it can be submitted to another journal. If any issues remain unresolved please contact the Publications Office at jphysiol@physoc.org

If you would like help with English language editing, or other article preparation support, Wiley Editing Services offers expert help with English Language Editing, as well as translation, manuscript formatting, and figure formatting at www.wileyauthors.com/eeo/preparation. You can also check out our resources for Preparing Your Article for general guidance about writing and preparing your manuscript at www.wileyauthors.com/eeo/prepresources.

Yours sincerely,

Richard Carson
Senior Editor
The Journal of Physiology

REQUIRED ITEMS:

Supplemental figures and tables are not allowed in the journal. Please could authors incorporate these into the manuscript (or else leave them out) as you see fit.

Please could authors include a Data Availability statement

EDITOR COMMENTS

Reviewing Editor:

The work by Soendenbroe et al. was evaluated by two experts in the field. Overall, the evaluation of the work was positive and deemed to be quite influential; however, both authors raised several concerns that should be addressed. These concerns prevent this manuscript from being accepted in its current form. The authors should address all of the reviewer's concerns in full, and believe that the changes to be made based on the reviewer comments will improve the quality and impact of the manuscript. Thank you for submitting your best work to the Journal of Physiology.

Senior Editor:

Please ensure that all relevant details (e.g., test statistic, degrees of freedom, exact probability etc) of the inferential statistical tests are presented clearly in the text (and that a statistical summary document is provided). There should be close adherence to the Statistical Policy of the Journal.

REFeree COMMENTS

Referee #1:

This study aimed to evaluate the health-promoting effects of lifelong physical activity by examining measures of physical function and muscle fiber physiology between groups of young, elderly lifelong exercise (LLEX), and elderly sedentary (SED) men. The major strengths of this study are a strong experimental design, clinically relevant findings, and the generation of novel information for the field. Overall, this is an important manuscript that is well-suited for The Journal of Physiology. However, in its current form, a number of minor concerns limit my enthusiasm for publication. Concerns are listed below.

The word "superior" in the title is misleading. Satellite cell content was higher in LLEX but the title makes it seem as if the function of those satellite cells was indeed greater than SED controls.

Introduction

The reference provided in the opening sentence is a good one, but the cohort study is exclusively Danish. Sarcopenia is a world-wide problem, do additional reports in less physically active countries support what is seen in Danes? If so, please add a citation to support the current one. If, indeed, the onset/rate of sarcopenia is different in other countries, this should be noted.

Skeletal muscle from older adults has several distinct phenotypes when compared to young controls. To claim that aged-muscle is characterized by denervation and reduced satellite cell number/function is a relatively narrow characterization. Language should be altered to more accurately reflect these are 2 of many phenotypes, or additional phenotypes described.

When the authors state "removes the specialization" what is meant by this? transcriptional specialization?

The last two sentences of the first paragraph are somewhat misleading. Satellite cells are required for myogenesis during pre- and post-natal skeletal muscle development and skeletal muscle regeneration throughout life. The complex nature of myogenesis should be made clear - the statement that the role of satellite cells during myogenesis is to "donate myonuclei" is inadequate. Further, while satellite cells are necessary for optimal skeletal muscle hypertrophy/adaptation they are not required for hypertrophy. This should be made explicitly clear and, as satellite cell measures are a critical piece of this manuscript, the relevant literature discussed (this could be done in the discussion or introduction). In particular, the mouse work that has examined the role of satellite cells in response to exercise-induced skeletal muscle adaptation (much of which has been completed by the Peterson group) is highly relevant to this project and should be discussed. Similarly, important NMJ/satellite cell literature is missing:

Larouche JA, Mohiuddin M, Choi JJ, Ulintz PJ, Fraczek P, Sabin K, Pitchiaya S, Kurpiers SJ, Castor-Macias J, Liu W, Hastings RL, Brown LA, Markworth JF, De Silva K, Levi B, Merajver SD, Valdez G, Chakkalakal JV, Jang YC, Brooks SV, Aguilar CA. Murine muscle stem cell response to perturbations of the neuromuscular junction are attenuated with aging. *Elife*. 2021 Jul 29;10:e66749. doi: 10.7554/eLife.66749. PMID: 34323217; PMCID: PMC8360658.

Liu W, Wei-LaPierre L, Klose A, Dirksen RT, Chakkalakal JV. Inducible depletion of adult skeletal muscle stem cells impairs the regeneration of neuromuscular junctions. *Elife*. 2015 Aug 27;4:e09221. doi: 10.7554/eLife.09221. PMID: 26312504; PMCID: PMC4579298.

The largest physical activity-based intervention completed to date is the LIFE study. This study should at least be

mentioned.

High levels of physical activity are associated with preserved muscle mass, strength, ect.

Higher lean body mass, not larger. When stating your hypotheses, please include the appropriate control group needed for comparison to test them.

The relevance of fiber-type specific changes with age needs to be mentioned somewhere prior to stating fiber-type specific hypotheses.

Methods

Please provide a rationalization for only recruiting men for this study and having a prior muscle biology as exclusion criteria.

The assessment of levels physical activity for appropriate group assignment is very qualitative. This is OK but needs to be stated as a limitation of the study.

Please explain what "sampling in opposite directions" means RE the biopsy procedure (may not require alterations to the text).

I am interested if the anatomical location of Pax7+/DAPI+ cells was considered in the quantification of satellite cell content.

Tables 1&2 can be supplementary

Results

I am unsure what "we observed a tendency" means. If a statistical trend was present, please explicitly say this and include the p-value. Also, please always include in which figures data are presented. These two issues appear in a few different sections of the results.

Figures need to be reorganized so they are introduced in a more linear/sequential fashion.

Fig. 3B needs a key.

Fig. 6 needs representative images across groups

Why did the authors choose to display with data in Fig 6A with line graphs?

How long after the exercise bout was the biopsy taken? Are the authors surprised they did not see an increase in SC content in response to exercise?

Referee #2:

The article by Soendenbroe and colleagues aimed to classify the impact of life long recreational exercise (LLEX) on skeletal muscle satellite cell quantity and function as well as innervation profile of type I and II muscle fibers. The authors collected muscle biopsy samples from life long recreational exercisers as well age matched sedentary controls and young subjects. Muscle function was assessed, muscle biopsy sections were analyzed as well as cultured myoblasts isolated from biopsy samples. The authors reported superior muscle function in LLEX as compared to sedentary and young subjects under challenged conditions. Additionally, LLEX had more satellite cells associated with Type II fibers as well as higher mRNA levels of acetylcholine receptors. The proportion of denervated fibers was not different between LLEX or sedentary and indices of satellite cell function, in vitro, was also not different between LLEX and age-matched sedentary subjects. The authors conclude that LLEX results in fatigue resistance and a more youthful satellite cell and acetylcholine receptor profile.

This study demonstrates that even a relatively small volume of recreational activity over a lifetime provides some level of protection against the aging skeletal muscle phenotype. Overall, this was a very well executed study with a very important message - even a recreational level of activity throughout life can help defend against some of the hallmark indices of skeletal muscle aging. Having said this, I do have some concerns about the interpretation of results and some of the approaches. I have outlined them below:

1. It is quite interesting that the authors report superior force output in the LEXX as compared to both young and sedentary individuals. This result surprised me and perhaps the authors as well. Given that the reduction in functional outcomes is typically attributed to "aging", per se, what does it mean that the LEXX outperformed the young? Is the described age-related loss in force a reflection of a sedentary lifestyle and not aging at all? Also, the authors make a compelling argument that the measures used in the study may not have been able to discriminate between recreationally active and sedentary individuals of similar age. However, one might expect differences to become apparent between LEXX and sedentary when comparing fibre type CSA, but this was not the case. Can the authors comment on this?

2. When assessing satellite cell "function", did the authors consider quantifying activated satellite cells in section or proliferation in vitro? The authors are relying on gene expression relating to myogenesis and innervation as measures of function, but this doesn't really inform on cell function per se. This part of the discussion should be restructured to accurately reflect what was measured. Also, measuring the expression of P16 alone as a marker of senescence falls short of being able to assess senescence. I would remove reference to this altogether.

3. I appreciate that the use of NCAM and MyHCn have previously been used to assess denervation, but the question remains whether this is actually an accurate measure of denervation status. Perhaps there could be some discussion regarding this interpretation. One might interpret small MyHCn fibers as remodelling or regenerating fibers.

4. There has been an attempt to link the loss of Type II fibre-associated satellite cells with type II fiber atrophy. There has been much discussion of what might come first. However, data from this study suggests that Type-II fiber associated satellite cells can be maintained in older adulthood while type II fibers atrophy. This suggests the two processes may not be linked at all. This may be worth highlighting.

END OF COMMENTS

Confidential Review

02-Dec-2021

Senior Editor:

Please ensure that all relevant details (e.g., test statistic, degrees of freedom, exact probability etc) of the inferential statistical tests are presented clearly in the text (and that a statistical summary document is provided). There should be close adherence to the Statistical Policy of the Journal.

Dear Senior Editor

In accordance with journal policy, we have changed the presentation of the summary data in *most* graphs/tables to average and standard deviations. The exceptions to this are:

- the skewed data sets, where non-parametric statistics were used, where we have retained the individual values with the median.
- The figures where we show inter-connected individual values for type I and II fibers, and averages (figure and 6.C and 8.A)
- Figure 7.C-D, where is not possible to show variation. Importantly, these data are not used for any statistical testing.
- For the creatine kinase and mRNA data, we show the geometric mean and 95% confidence intervals

All of the above is carefully described in the legends of the specific figure/table.

We have also provided exact p values throughout the results section and in table 4, as well as adding a statistical summary document.

Reviewing editor:

The work by Soendenbroe et al. was evaluated by two experts in the field. Overall, the evaluation of the work was positive and deemed to be quite influential; however, both authors raised several concerns that should be addressed. These concerns prevent this manuscript from being accepted in its current form. The authors should address all of the reviewer's concerns in full, and believe that the changes to be made based on the reviewer comments will improve the quality and impact of the manuscript. Thank you for submitting your best work to the Journal of Physiology.

Dear Reviewing Editor

We have addressed all points brought up by the two referees, and we hope that the adapted version of the manuscript will be acceptable to all.

Additionally, we would like to point out a mistake that was discovered during the preparation of the revised figures and the statistical summary document. In the original supplemental material 8 (Figure 10 in the revised version), the significant exercise responses in MyHCe and AChR $\alpha 1$ (tendency) at the tissue level had been assigned to the SED group, when it should have been the LLEX group. This has been corrected in the revised graph and the associated section of the results. The revised sentence is found on page 3, line 430-431, in the version with tracked changes: "MyHCe gene expression was upregulated in the exercised leg of LLEX ($p=0.035$) ...". These data were not used in the discussion, so no further changes have been made.

Referee #1:

This study aimed to evaluate the health-promoting effects of lifelong physical activity by examining measures of physical function and muscle fiber physiology between groups of young, elderly lifelong exercise (LLEX), and elderly sedentary (SED) men. The major strengths of this study are a strong experimental design, clinically relevant findings, and the generation of novel information for the field. Overall, this is an important manuscript that is well-suited for The Journal of Physiology. However, in its current form, a number of minor concerns limit my enthusiasm for publication. Concerns are listed below.

We thank the reviewer for the kind words and the many relevant corrections and suggestions. We have tried to accommodate all of them as well as possible.

The word "superior" in the title is misleading. Satellite cell content was higher in LLEX but the title makes it seem as if the function of those satellite cells was indeed greater than SED controls.

We agree the title is misleading, and we have therefore changed the word "Superior" to "Preserved". The new title is: "Preserved stem cell content and innervation profile of elderly human skeletal muscle with lifelong recreational exercise"

Introduction

The reference provided in the opening sentence is a good one, but the cohort study is exclusively Danish. Sarcopenia is a world-wide problem, do additional reports in less physically active countries support what is seen in Danes? If so, please add a citation to support the current one. If, indeed, the onset/rate of sarcopenia is different in other countries, this should be noted.

This is a good point, especially as Danes, at least in some geographical areas, are generally more leisurely active compared to other western countries.

We have added two studies which we believe are in agreement with the study by Suetta et al., 2019. The first study is by Janssen et al., 2000, where muscle mass was assessed in 268 men and 200 women aged between 18-88, in Canada and USA. The second study is by Kostka, 2005 who evaluated muscle mass and function (quadriceps power on cycle ergometer) in 335 men aged between 23-88 years, in Poland.

Kostka T (2005). Quadriceps maximal power and optimal shortening velocity in 335 men aged 23–88 years. *Eur J Appl Physiol* 95, 140–145

Janssen I, Heymsfield SB, Wang Z & Ross R (2000). Skeletal muscle mass and distribution in 468 men and women aged 18–88 yr. *Journal of Applied Physiology* 89, 81–88

Skeletal muscle from older adults has several distinct phenotypes when compared to young controls. To claim that aged-muscle is characterized by denervation and reduced satellite cell number/function is a relatively narrow characterization. Language should be altered to more accurately reflect these are 2 of many phenotypes, or additional phenotypes described.

Agreed. The new sentence is found on page 3, line 70-72, in the version with tracked changes: "Among the myriad of changes associated with the ageing muscle, myofibre denervation and a decline in the number and function of muscle stem (satellite) cells are clear features"

When the authors state "removes the specialization" what is meant by this? transcriptional specialization?

Yes, indeed transcriptional specialization. Agreed. The new sentence is found on page 3, line 77-79, in the version with tracked changes: “Loss of myofibre innervation removes the transcriptional specialization normally confined to the small synaptic area, and alters gene expression in the extra-synaptic area of the myofibre”.

The last two sentences of the first paragraph are somewhat misleading. Satellite cells are required for myogenesis during pre- and post-natal skeletal muscle development and skeletal muscle regeneration throughout life. The complex nature of myogenesis should be made clear - the statement that the role of satellite cells during myogenesis is to "donate myonuclei" is inadequate. Further, while satellite cells are necessary for optimal skeletal muscle hypertrophy/adaptation they are not required for hypertrophy. This should be made explicitly clear and, as satellite cell measures are a critical piece of this manuscript, the relevant literature discussed (this could be done in the discussion or introduction). In particular, the mouse work that has examined the role of satellite cells in response to exercise-induced skeletal muscle adaptation (much of which has been completed by the Peterson group) is highly relevant to this project and should be discussed. Similarly, important NMJ/satellite cell literature is missing:

Larouche JA, Mohiuddin M, Choi JJ, Ulintz PJ, Fraczek P, Sabin K, Pitchiaya S, Kurpiers SJ, Castor-Macias J, Liu W, Hastings RL, Brown LA, Markworth JF, De Silva K, Levi B, Merajver SD, Valdez G, Chakkalakal JV, Jang YC, Brooks SV, Aguilar CA. Murine muscle stem cell response to perturbations of the neuromuscular junction are attenuated with aging. *Elife*. 2021 Jul 29;10:e66749. doi: 10.7554/eLife.66749. PMID: 34323217; PMCID: PMC8360658.

Liu W, Wei-LaPierre L, Klose A, Dirksen RT, Chakkalakal JV. Inducible depletion of adult skeletal muscle stem cells impairs the regeneration of neuromuscular junctions. *Elife*. 2015 Aug 27;4:e09221. doi: 10.7554/eLife.09221. PMID: 26312504; PMCID: PMC4579298.

We agree and have completely rewritten our introduction of SCs. See page 3-4: “Satellite cells are indispensable during embryonic myogenesis and for muscle regeneration during adulthood (Engquist & Zammit, 2021), due to their ability to proliferate, fuse and form myotubes. Given their role as the sole source of myonuclei, satellite cells are also involved in the hypertrophic response to exercise (Murach *et al.*, 2021a). Studies using satellite cell depleted mice have shown that some hypertrophy can be achieved without satellite cells, but in order to maximize the response to long-term training, satellite cells are required (Englund *et al.*, 2020). It is now also clear that satellite cells interact directly with muscle fibers (Murach *et al.*, 2021b) and with other cell types located in the microenvironment surrounding the muscle fiber, including fibroblasts (Mackey *et al.*, 2017; Fry *et al.*, 2017) and endothelial cells (Nederveen *et al.*, 2021). Maladaptation of the muscle is evident during persistent overload in the absence of satellite cells, such as increased ECM and fibroblast number, indicating a regulatory role for satellite cells in ameliorating unfavourable remodelling of the muscle environment (Murach *et al.*, 2018). In relation to the NMJ, it has been shown that a subgroup of satellite cells generate and maintain the specialized myonuclei at the NMJ (Liu *et al.*, 2017; Larouche *et al.*, 2021) and that depletion of satellite cells dampens the regeneration of NMJs following nerve damage (Liu *et al.*, 2015). Although not completely depleted, the aged human muscle has been shown to have fewer satellite cells, especially associated with type II fibres (Verdijk *et al.*, 2014; Karlsen *et al.*, 2019, 2020). Furthermore, a link between denervation and satellite cells has been shown, where satellite cells exit the quiescent state following denervation and mount an attempt at compensatory myogenesis (Borisov *et al.*, 2001). Long-term denervated fibers also possess viable satellite cells with preserved renewal capability (Wong *et al.*, 2021).

The largest physical activity-based intervention completed to date is the LIFE study. This study should at least be mentioned.

High levels of physical activity are associated with preserved muscle mass, strength, ect.

We agree, and have included Pahor et al., 2020, a review describing the overall findings of the LIFE study, as a reference when introducing exercise as a tool to modulate age-related changes in muscle function.

Pahor M, Guralnik JM, Anton SD, Ambrosius WT, Blair SN, Church TS, Espeland MA, Fielding RA, Gill TM, Glynn NW, Groessl EJ, King AC, Kritchevsky SB, Manini TM, McDermott MM, Miller ME, Newman AB & Williamson JD (2020). Impact and Lessons From the Lifestyle Interventions and Independence for Elders (LIFE) Clinical Trials of Physical Activity to Prevent Mobility Disability. *J Am Geriatr Soc* 68, 872–881.

Higher lean body mass, not larger. When stating your hypotheses, please include the appropriate control group needed for comparison to test them.

Amended.

The new sentence is found on page 4-5, line 131-137, in the version with tracked changes: “We hypothesized that physically active individuals would possess a higher lean body mass and better muscle function compared to sedentary individuals, although an inherent decline in muscle morphology and function due to ageing would still exist (relative to young control group). Furthermore, we hypothesized that positive effects of lifelong recreational physical activity would be evident for indices of myofibre denervation, myofibre size, type II myofibre associated satellite cells, and satellite cell function in cell culture in comparison to a sedentary lifestyle.”

The relevance of fiber-type specific changes with age needs to be mentioned somewhere prior to stating fiber-type specific hypotheses.

Agree, amended.

The new sentence is found on page 3, line 98-99, in the version with tracked changes: “Although not completely depleted, the aged human muscle has been shown to have fewer satellite cells, especially associated with type II fibres (Verdijk *et al.*, 2014; Karlsen *et al.*, 2019, 2020).”

Methods

Please provide a rationalization for only recruiting men for this study and having a prior muscle biology as exclusion criteria.

As the study contained in vivo work, cell cultures, gene expression analysis and immunofluorescence, we wanted to increase the homogeneity of our subjects, as we would otherwise have to increase the number of subjects to be able to detect significant differences. Some of the measures were also quite exploratory in nature, meaning that we did not know what magnitude of difference we could expect. It is clear that this decision limits the scope of the findings, as they only relate to approximately half of the world’s population, and we have added this limitation to the discussion, encouraging additional studies in females to follow up on our findings.

The new sentence is found on page 20, line 585-588, in the version with tracked changes: “The study is limited by the lack of objective measures of levels of physical activity and the inclusion of only male participants. In our earlier study on young and elderly females, similar findings on myofiber denervation were reported. Clearly, studies of lifelong exercise in females are needed.”

In relation to a prior biopsy as an exclusion criterion, we wanted to be certain our findings were not confounded by previous biopsy sampling, which potentially would be characterised by scar tissue.

The assessment of levels physical activity for appropriate group assignment is very qualitative. This is OK but needs to be stated as a limitation of the study.

We agree that a more standardized and objective approach would have been better.

We have added a sentence in relation to a recently published classification framework by McKay et al., 2022, which place the subjects of the active group in Tier 1 (recreationally active). These individuals meet the recommendations for physical activity set by the World Health Organization, often through a combination of different activities, and without a specific aim at competing.

McKay AKA, Stellingwerff T, Smith ES, Martin DT, Mujika I, Goosey-Tolfrey VL, Sheppard J & Burke LM (2022). Defining Training and Performance Caliber: A Participant Classification Framework. *Int J Sports Physiol Perform* 1–15.

The new sentence is found on page 20, line 585-586, in the version with tracked changes: “The study is limited by the lack of objective measures of levels of physical activity and the inclusion of only male participants.”

Please explain what "sampling in opposite directions" means RE the biopsy procedure (may not require alterations to the text).

We have made changes to clarify this. The new sentence is found on page 8, line 208-210, in the version with tracked changes: “Two biopsies were taken from each leg in immediate succession, through the same incision, with the biopsy needle angled proximally and distally from the incision”

I am interested if the anatomical location of Pax7+/DAPI+ cells was considered in the quantification of satellite cell content.

Laminin was used as a marker for the basement membrane, which was used, together with Pax7 and Dapi, to identify satellite cells. Pax7+ DAPI+ cells located at the periphery of myofibre were classified as satellite cells. In most cases the satellite cell was also clearly located within the myofibre basement membrane (laminin). 14 satellite cells could not be assigned to a particular fibre but the uncertainty was mostly due to tissue section imperfections at that specific site, rather than the satellite cell being situated outside the basement membrane. During our analysis we did not address the anatomical location of satellite cells in other ways or make any attempts to assess it descriptively. It is a good point though, as recent studies have shed light on the importance of satellite cell proximity to capillaries (Nederveen *et al.*, 2021) as well as the finding of “incarcerated” satellite cells, completely surrounded by laminin (Nederveen *et al.*, 2020). In the case of investigating the latter, confocal imaging would be required.

Tables 1&2 can be supplementary

Yes we agree these tables are supplementary in nature, however since JoP does not allow supplementary files we have to include them in the main manuscript.

Results

I am unsure what "we observed a tendency" means. If a statistical trend was present, please explicitly say this and include the p-value. Also, please always include in which figures data are presented. These two

issues appear in a few different sections of the results.

Agree, this was unclear. As per journal policy, we have added specific p values and figure references for the results presented.

Figures need to be reorganized so they are introduced in a more linear/sequential fashion.

We have reordered the tables and figures to reflect the order of appearance. All “supplemental” material has also been incorporated into the main manuscript, in line with JP requirements.

Fig. 3B needs a key.

The figure (now 5.A) has been merged with supplemental figure 2, and significance symbols inserted appropriately.

We agree that the figure is complex and most likely can only be interpreted together with figure 1.B. The overall message is 1) force is gradually diminished during exercise and 2) exercise habituated individuals perform better than sedentary individuals.

Fig. 6 needs representative images across groups

We have inserted three representative images, one for each group (now 8.C).

Why did the authors choose to display with data in Fig 6A with line graphs?

We felt it was important to link type I and II fiber data from each subject in a clear way. We have done the same for fibre CSA in figure 4.C (now 6C).

How long after the exercise bout was the biopsy taken? Are the authors surprised they did not see an increase in SC content in response to exercise?

The biopsies were obtained 6 days after the exercise bout, and it was indeed a surprise that no increase in satellite cell content was observed. This is also in contrast to the majority of studies on the topic, as increases in satellite cells following acute exercise have been reported, especially following eccentric contractions (Hyldahl *et al.*, 2014; Heisterberg *et al.*, 2018). It might be that the 6 day time point was too late, and the peak in satellite cell proliferation was missed (Snijders *et al.*, 2015). All contractions were performed at a slow tempo (30 degree per second for 80-degree range of motion), which is slower than what is used in most studies. It could also be simply that the exercise bout was not hard enough. The latter explanation seems unlikely, due the marked decrease in muscle strength following the exercise bout and the significant increase in creatine kinase levels in all groups.

We have added the following sentence in the discussion on satellite cells: “Surprisingly, the exercise bout did not lead to an increase in satellite cell content, which might be related to the exclusive use of slow contractions, timing of biopsy sampling or insufficient stimulus (Hyldahl & Hubal, 2014; Snijders *et al.*, 2015).”

Referee #2:

The article by Soendenbroe and colleagues aimed to classify the impact of life long recreational exercise (LLEX) on skeletal muscle satellite cell quantity and function as well as innervation profile of type I and II muscle fibers. The authors collected muscle biopsy samples from life long recreational exercisers as well as age matched sedentary controls and young subjects. Muscle function was assessed, muscle biopsy sections were analyzed as well as cultured myoblasts isolated from biopsy samples. The authors reported superior muscle function in LLEX as compared to sedentary and young subjects under challenged conditions. Additionally, LLEX had more satellite cells associated with Type II fibers as well as higher mRNA levels of acetylcholine receptors. The proportion of denervated fibers was not different between LLEX or sedentary and indices of satellite cell function, in vitro, was also not different between LLEX and age-matched sedentary subjects. The authors conclude that LLEX results in fatigue resistance and a more youthful satellite cell and acetylcholine receptor profile. This study demonstrates that even a relatively small volume of recreational activity over a lifetime provides some level of protection against the aging skeletal muscle phenotype. Overall, this was a very well executed study with a very important message - even a recreational level of activity throughout life can help defend against some of the hallmark indices of skeletal muscle aging. Having said this, I do have some concerns about the interpretation of results and some of the approaches. I have outlined them below:

We sincerely thank the reviewer for these kind words. We have answered the specific comments and made the appropriate changes in the manuscript.

1. It is quite interesting that the authors report superior force output in the LEXX as compared to both young and sedentary individuals. This result surprised me and perhaps the authors as well. Given that the reduction in functional outcomes is typically attributed to "aging", per se, what does it mean that the LEXX outperformed the young? Is the described age-related loss in force a reflection of a sedentary lifestyle and not aging at all?

This was indeed a very surprising result, and the idea is interesting that it might not be ageing per se, but rather the wide range in activity levels which determines muscle function during ageing – at least to some extent. Studies of “elite” master athletes have revealed that even individuals operating at the pinnacle of sporting performance, within their age-group, still experience a gradual decline in muscle function with age (Lazarus & Harridge, 2017).

The difference between LLEX and SED in the performance during the exercise bout most likely reflects a true physiological phenomenon. The underlying physiological mechanisms were not studied in the present study and are likely to be wide-ranging from local metabolic, to neural. The fact that this is not known, and that the group of recreationally active individuals constitute such a large part of the population relative to master athletes, points to a gaping hole in the literature.

Also, the authors make a compelling argument that the measures used in the study may not have been able to discriminate between recreationally active and sedentary individuals of similar age. However, one might expect differences to become apparent between LEXX and sedentary when comparing fibre type CSA, but this was not the case. Can the authors comment on this?

Muscle fiber CSA is reduced with ageing, and we initially thought that the subjects in LLEX might be protected from this decline, when compared with SED. However, this was not the case. If we search the literature for an explanation it quickly becomes clear that the findings on fiber CSA and activity level are largely inconclusive (Klitgaard *et al.*, 1990; Mackey *et al.*, 2014; Zampieri *et al.*, 2015; St-Jean-Pelletier *et al.*, 2017; Sonjak *et al.*, 2019; McKendry *et al.*, 2020). Mackey *et al.*, 2014 reports similar fiber type specific CSA in lifelong runners and sedentary matched controls, which could indicate that the endurance type activities do not provide a sufficient stimulus to preserve muscle size. On the other hand, Sonjak *et al.*, 2019

compared master athletes within track&field to a group of pre-frail/frail elderly individuals and also found no difference in fiber CSA. Others find protective effects of physical activity on fiber size (Klitgaard *et al.*, 1990; Zampieri *et al.*, 2015). One important consideration is that the SED group of the present study had to be matched to the LLEX group on BMI, and could not be obese, take certain kinds of medicine, drink too much alcohol, etc. So, the SED group might be “artificially” healthy and perhaps in some way protected from an age-related decline.

2. When assessing satellite cell "function", did the authors consider quantifying activated satellite cells in section or proliferation in vitro?

We did intend to study proliferation in vitro using the BrdU approach, as we have done in prior studies (Mackey *et al.*, 2017; Bechshøft *et al.*, 2019). However, the staining was unsuccessful, so the analysis could not be conducted in a reliable manner. We agree with the reviewer that this aspect of satellite cell activity is equally important and interesting compared with differentiation and fusion and might have provided some valuable information.

The new sentence is found on page 17, line 487-489, in the version with tracked changes: “Satellite cell proliferation could not be assessed due to problems relating to the staining protocol, so we cannot rule out potential differences between groups in myoblast proliferation.”

The authors are relying on gene expression relating to myogenesis and innervation as measures of function, but this doesn't really inform on cell function per se. This part of the discussion should be restructured to accurately reflect what was measured.

In relation to the comment on satellite cell function, we agree that we put too much emphasis on the gene expression data. We do however also believe that the gene expression data are related to cellular function, and therefore can be discussed in that regard. But we have made changes to make it clear that the key data on cellular function in our study are the differentiation and fusion index, and that they are supported by gene expression data.

The new sentences are found on page 17, line 484-487 and 504-505, in the version with tracked changes: “Contrary to our hypothesis, the two primary measures of cell function, differentiation and fusion index, were similar in LLEX and SED, while only a tendency for an age-related difference for fusion index was observed, which might be explained by a higher cell number.” AND “However, neither differentiation nor fusion index, the primary measures of cell function, were affected by age, although the influence on proliferation remains to be determined.”

Also, measuring the expression of P16 alone as a marker of senescence falls short of being able to assess senescence. I would remove reference to this altogether.

We agree that the use of p16 alone is an unsatisfactory assessment of cellular senescence, and we have removed the word senescence.

3. I appreciate that the use of NCAM and MyHCn have previously been used to assess denervation, but the question remains whether this is actually an accurate measure of denervation status. Perhaps there could be some discussion regarding this interpretation. One might interpret small MyHCn fibers as remodelling or regenerating fibers.

We agree the NCAM and MyHCn are ambiguous markers to some extent, being found at NMJs (Moore & Walsh, 1985; Vaitinen *et al.*, 1999), MTJs (Jakobsen *et al.*, 2018), in intrafusal fibers (Walro & Kucera, 1999;

Schiaffino *et al.*, 2015), specific facial muscles (Sartore *et al.*, 1987; Butler-Browne *et al.*, 1988; Stål *et al.*, 1994), as well as in diseased (Walsh & Moore, 1985; Winter & Bornemann, 1999; Gosztonyi *et al.*, 2001; Doppler *et al.*, 2008; Daou *et al.*, 2020), developing (Butler-Browne & Whalen, 1984; Moore & Walsh, 1985) and regenerating (Schiaffino *et al.*, 1988; Irintchev *et al.*, 1994; Mackey & Kjaer, 2017) muscle. Myogenic cell cultures also express NCAM and MyHCn. In a recent preprint uploaded to bioRxiv, Lin *et al.*, 2021 reported increased NCAM transcription, using single-nucleus RNAseq, in denervated gastrocnemius mouse muscle (Lin *et al.*, 2021), and there are a number of denervation studies (Butler-Browne *et al.*, 1982; Covault & Sanes, 1985; Covault *et al.*, 1986; Schiaffino *et al.*, 1988; Olsen *et al.*, 1995; Xing *et al.*, 2015) that show clear increases in both NCAM and MyHCn expression following denervation. We believe that denervation is the most likely explanation for the increased number of NCAM+ and MyHCn+ fibers, although, with the limited palette of markers currently available, we cannot be certain. To reflect this uncertainty, we have adjusted the text in the paragraph before outlining the primary findings in the discussion. The new sentence is found on page 18, line 515-525, in the version with tracked changes: “It should be noted that NCAM and MyHCn are also associated with other physiological processes and structures within muscle, which can challenge the interpretation. NCAM is found at the NMJ and MTJ (Moore & Walsh, 1985; Jakobsen *et al.*, 2018), during muscle regeneration (Irintchev *et al.*, 1994; Mackey & Kjaer, 2017) and in neuromuscular disease (Walsh & Moore, 1985). MyHCn is found during muscle regeneration (Sartore *et al.*, 1982; Mackey & Kjaer, 2017), in neuromuscular disease (Fitzsimons & Hoh, 1981) and in intrafusal fibres (Walro & Kucera, 1999). However, in healthy vastus lateralis muscle tissue, MTJ and NMJ structures are easily recognized, intrafusal fibers are rare, and muscle regeneration is unlikely to be present. Furthermore, experimentally-induced muscle denervation leads to a large upregulation in the expression of NCAM and MyHCn (Covault & Sanes, 1985; Schiaffino *et al.*, 1988), together making muscle fibre denervation the most likely explanation for the observation of NCAM⁺ and MyHCn⁺ fibres in our study.”

4. There has been an attempt to link the loss of Type II fibre-associated satellite cells with type II fiber atrophy. There has been much discussion of what might come first. However, data from this study suggests that Type-II fiber associated satellite cells can be maintained in older adulthood while type II fibers atrophy. This suggests the two processes may not be linked at all. This may be worth highlighting.

This is a very good point. A substantial number of studies have shown that fibre atrophy, especially for type II fibres, occurs with ageing, and much has been done to find the mechanisms involved. One idea is that satellite cells, perhaps through a need for constant supply of myonuclei, are involved in the maintenance of myofibers. Verdijk *et al.*, 2007 provided evidence that type II fiber atrophy was accompanied by a lower type II fibre associated satellite cell count, which was interpreted as a causal link between the two (Verdijk *et al.*, 2007). The same group have also provided evidence for the reverse process, that is a concomitant increase in fiber size and satellite cell numbers following prolonged resistance exercise, as discussed here (Snijders *et al.*, 2009). Given that the subjects in the present study have been training for many years prior to the point of investigation, it is unlikely that they currently experience myofibre hypertrophy. From this it can then be argued that changes in fiber size comes first, with ensuing changes in satellite cell content. Alternatively, these two entities might not be as closely related as previously thought.

We have inserted a sentence on this topic into the discussion of fibre CSA. The new sentence is found on page 19, line 566-569, in the version with tracked changes: “It is also noteworthy that while the amount and type of activity performed by LLEX did not appear to preserve type II myofibre size, it was associated with a preservation of the number of type II myofibre associated satellite cells, suggesting that these two entities are not tightly regulated in healthy elderly muscle.”

References

- Bechshøft CJL, Jensen SM, Schjerling P, Andersen JL, Svensson RB, Eriksen CS, Mkumbuzi NS, Kjaer M & Mackey AL (2019). Age and prior exercise in vivo determine the subsequent in vitro molecular profile of myoblasts and nonmyogenic cells derived from human skeletal muscle. *Am J Physiol, Cell Physiol* **316**, C898–C912.
- Borisov AB, Dedkov EI & Carlson BM (2001). Interrelations of myogenic response, progressive atrophy of muscle fibers, and cell death in denervated skeletal muscle. *Anat Rec* **264**, 203–218.
- Butler-Browne GS, Bugaisky LB, Cuénoud S, Schwartz K & Whalen RG (1982). Denervation of newborn rat muscle does not block the appearance of adult fast myosin heavy chain. *Nature* **299**, 830–833.
- Butler-Browne GS, Eriksson PO, Laurent C & Thornell LE (1988). Adult human masseter muscle fibers express myosin isozymes characteristic of development. *Muscle Nerve* **11**, 610–620.
- Butler-Browne GS & Whalen RG (1984). Myosin isozyme transitions occurring during the postnatal development of the rat soleus muscle. *Dev Biol* **102**, 324–334.
- Covault J, Merlie JP, Goridis C & Sanes JR (1986). Molecular forms of N-CAM and its RNA in developing and denervated skeletal muscle. *J Cell Biol* **102**, 731–739.
- Covault J & Sanes JR (1985). Neural cell adhesion molecule (N-CAM) accumulates in denervated and paralyzed skeletal muscles. *Proceedings of the National Academy of Sciences* **82**, 4544–4548.
- Daou N, Hassani M, Matos E, De Castro GS, Galvao Figueredo Costa R, Seelaender M, Moresi V, Rocchi M, Adamo S, Li Z, Agbulut O & Coletti D (2020). Displaced Myonuclei in Cancer Cachexia Suggest Altered Innervation. *International Journal of Molecular Sciences* **21**, 1092.
- Doppler K, Mittelbronn M & Bornemann A (2008). Myogenesis in human denervated muscle biopsies. *Muscle & Nerve* **37**, 79–83.
- Englund DA, Murach KA, Dungan CM, Figueiredo VC, Vechetti IJ, Dupont-Versteegden EE, McCarthy JJ & Peterson CA (2020). Depletion of resident muscle stem cells negatively impacts running volume, physical function, and muscle fiber hypertrophy in response to lifelong physical activity. *Am J Physiol Cell Physiol* **318**, C1178–C1188.
- Engquist EN & Zammit PS (2021). The Satellite Cell at 60: The Foundation Years. *J Neuromuscul Dis* **8**, S183–S203.
- Fitzsimons RB & Hoh JF (1981). Embryonic and foetal myosins in human skeletal muscle. The presence of foetal myosins in duchenne muscular dystrophy and infantile spinal muscular atrophy. *J Neurol Sci* **52**, 367–384.
- Fry CS, Kirby TJ, Kosmac K, McCarthy JJ & Peterson CA (2017). Myogenic Progenitor Cells Control Extracellular Matrix Production by Fibroblasts during Skeletal Muscle Hypertrophy. *Cell Stem Cell* **20**, 56–69.
- Gosztonyi G, Naschold U, Grozdanovic Z, Stoltenburg-Didinger G & Gossrau R (2001). Expression of Leu-19 (CD56, N-CAM) and nitric oxide synthase (NOS) I in denervated and reinnervated human skeletal muscle. *Microscopy Research and Technique* **55**, 187–197.

- Heisterberg MF, Andersen JL, Schjerling P, Bülow J, Lauersen JB, Roeber HL, Kjaer M & Mackey AL (2018). Effect of Losartan on the Acute Response of Human Elderly Skeletal Muscle to Exercise. *Med Sci Sports Exerc* **50**, 225–235.
- Hyldahl RD & Hubal MJ (2014). Lengthening our perspective: Morphological, cellular, and molecular responses to eccentric exercise. *Muscle & Nerve* **49**, 155–170.
- Hyldahl RD, Olson T, Welling T, Groskost L & Parcell AC (2014). Satellite cell activity is differentially affected by contraction mode in human muscle following a work-matched bout of exercise. *Front Physiol* **5**, 485.
- Irintchev A, Zeschnigk M, Starzinski-Powitz A & Wernig A (1994). Expression pattern of M-cadherin in normal, denervated, and regenerating mouse muscles. *Dev Dyn* **199**, 326–337.
- Jakobsen JR, Jakobsen NR, Mackey AL, Koch M, Kjaer M & Krogsgaard MR (2018). Remodeling of muscle fibers approaching the human myotendinous junction. *Scandinavian Journal of Medicine & Science in Sports* **28**, 1859–1865.
- Karlsen A, Bechshøft RL, Malmgaard-Clausen NM, Andersen JL, Schjerling P, Kjaer M & Mackey AL (2019). Lack of muscle fibre hypertrophy, myonuclear addition, and satellite cell pool expansion with resistance training in 83-94-year-old men and women. *Acta Physiol (Oxf)* **227**, e13271.
- Karlsen A, Soendenbroe C, Malmgaard-Clausen NM, Wagener F, Moeller CE, Senhaji Z, Damberg K, Andersen JL, Schjerling P, Kjaer M & Mackey AL (2020). Preserved capacity for satellite cell proliferation, regeneration, and hypertrophy in the skeletal muscle of healthy elderly men. *FASEB J* **34**, 6418–6436.
- Klitgaard H, Mantoni M, Schiaffino S, Ausoni S, Gorza L, Laurent-Winter C, Schnohr P & Saltin B (1990). Function, morphology and protein expression of ageing skeletal muscle: a cross-sectional study of elderly men with different training backgrounds. *Acta Physiol Scand* **140**, 41–54.
- Larouche JA et al. (2021). Murine muscle stem cell response to perturbations of the neuromuscular junction are attenuated with aging. *Elife* **10**, e66749.
- Lazarus NR & Harridge SDR (2017). Declining performance of master athletes: silhouettes of the trajectory of healthy human ageing? *J Physiol (Lond)* **595**, 2941–2948.
- Lin H, Ma X, Sun Y, Peng H, Wang Y, Thomas SS & Hu Z (2021). Decoding the transcriptome of denervated muscle at single-nucleus resolution. *bioRxiv*2021.10.25.463678.
- Liu W, Klose A, Forman S, Paris ND, Wei-LaPierre L, Cortés-Lopéz M, Tan A, Flaherty M, Miura P, Dirksen RT & Chakkalakal JV (2017). Loss of adult skeletal muscle stem cells drives age-related neuromuscular junction degeneration ed. Wagers AJ. *eLife* **6**, e26464.
- Liu W, Wei-LaPierre L, Klose A, Dirksen RT & Chakkalakal JV (2015). Inducible depletion of adult skeletal muscle stem cells impairs the regeneration of neuromuscular junctions. *eLife* **4**, e09221.
- Mackey AL, Karlsen A, Couppé C, Mikkelsen UR, Nielsen RH, Magnusson SP & Kjaer M (2014). Differential satellite cell density of type I and II fibres with lifelong endurance running in old men. *Acta Physiologica* **210**, 612–627.

- Mackey AL & Kjaer M (2017). The breaking and making of healthy adult human skeletal muscle in vivo. *Skeletal Muscle* **7**, 24.
- Mackey AL, Magnan M, Chazaud B & Kjaer M (2017). Human skeletal muscle fibroblasts stimulate in vitro myogenesis and in vivo muscle regeneration. *J Physiol (Lond)* **595**, 5115–5127.
- McKendry J, Joanisse S, Baig S, Liu B, Parise G, Greig CA & Breen L (2020). Superior Aerobic Capacity and Indices of Skeletal Muscle Morphology in Chronically Trained Master Endurance Athletes Compared With Untrained Older Adults. *J Gerontol A Biol Sci Med Sci* **75**, 1079–1088.
- Moore SE & Walsh FS (1985). Specific regulation of N-CAM/D2-CAM cell adhesion molecule during skeletal muscle development. *EMBO J* **4**, 623–630.
- Murach KA, Fry CS, Dupont-Versteegden EE, McCarthy JJ & Peterson CA (2021a). Fusion and beyond: Satellite cell contributions to loading-induced skeletal muscle adaptation. *FASEB J* **35**, e21893.
- Murach KA, Fry CS, Kirby TJ, Jackson JR, Lee JD, White SH, Dupont-Versteegden EE, McCarthy JJ & Peterson CA (2018). Starring or Supporting Role? Satellite Cells and Skeletal Muscle Fiber Size Regulation. *Physiology (Bethesda)* **33**, 26–38.
- Murach KA, Peck BD, Policastro RA, Vechetti IJ, Van Pelt DW, Dungan CM, Denes LT, Fu X, Brightwell CR, Zentner GE, Dupont-Versteegden EE, Richards CI, Smith JJ, Fry CS, McCarthy JJ & Peterson CA (2021b). Early satellite cell communication creates a permissive environment for long-term muscle growth. *iScience* **24**, 102372.
- Nederveen JP, Betz MW, Snijders T & Parise G (2021). The Importance of Muscle Capillarization for Optimizing Satellite Cell Plasticity. *Exerc Sport Sci Rev* **49**, 284–290.
- Nederveen JP, Joanisse S, Thomas ACQ, Snijders T, Manta K, Bell KE, Phillips SM, Kumbhare D & Parise G (2020). Age-related changes to the satellite cell niche are associated with reduced activation following exercise. *FASEB J* **34**, 8975–8989.
- Olsen M, Zuber C, Roth J, Linnemann D & Bock E (1995). The ability to re-express polysialylated NCAM in soleus muscle after denervation is reduced in aged rats compared to young adult rats. *International Journal of Developmental Neuroscience* **13**, 97–104.
- Sartore S, Gorza L & Schiaffino S (1982). Fetal myosin heavy chains in regenerating muscle. *Nature* **298**, 294–296.
- Sartore S, Mascarello F, Rowleron A, Gorza L, Ausoni S, Vianello M & Schiaffino S (1987). Fibre types in extraocular muscles: a new myosin isoform in the fast fibres. *J Muscle Res Cell Motil* **8**, 161–172.
- Schiaffino S, Gorza L, Pitton G, Saggin L, Ausoni S, Sartore S & Lømo T (1988). Embryonic and neonatal myosin heavy chain in denervated and paralyzed rat skeletal muscle. *Developmental Biology* **127**, 1–11.
- Schiaffino S, Rossi AC, Smerdu V, Leinwand LA & Reggiani C (2015). Developmental myosins: expression patterns and functional significance. *Skeletal Muscle*; DOI: 10.1186/s13395-015-0046-6.
- Snijders T, Nederveen JP, McKay BR, Joanisse S, Verdijk LB, van Loon LJC & Parise G (2015). Satellite cells in human skeletal muscle plasticity. *Front Physiol* **6**, 283.

- Snijders T, Verdijk LB & van Loon LucJC (2009). The impact of sarcopenia and exercise training on skeletal muscle satellite cells. *Ageing Research Reviews* **8**, 328–338.
- Sonjak V, Jacob K, Morais JA, Rivera-Zengotita M, Spendiff S, Spake C, Taivassalo T, Chevalier S & Hepple RT (2019). Fidelity of muscle fibre reinnervation modulates ageing muscle impact in elderly women. *The Journal of Physiology* **597**, 5009–5023.
- Stål P, Eriksson PO, Schiaffino S, Butler-Browne GS & Thornell LE (1994). Differences in myosin composition between human oro-facial, masticatory and limb muscles: enzyme-, immunohisto- and biochemical studies. *J Muscle Res Cell Motil* **15**, 517–534.
- St-Jean-Pelletier F, Pion CH, Leduc-Gaudet J-P, Sgaroto N, Zovilé I, Barbat-Artigas S, Reynaud O, Alkaterji F, Lemieux FC, Grenon A, Gaudreau P, Hepple RT, Chevalier S, Belanger M, Morais JA, Aubertin-Leheudre M & Gousspillou G (2017). The impact of ageing, physical activity, and pre-frailty on skeletal muscle phenotype, mitochondrial content, and intramyocellular lipids in men. *Journal of Cachexia, Sarcopenia and Muscle* **8**, 213–228.
- Vaittinen S, Lukka R, Sahlgren C, Rantanen J, Hurme T, Lendahl U, Eriksson JE & Kalimo H (1999). Specific and innervation-regulated expression of the intermediate filament protein nestin at neuromuscular and myotendinous junctions in skeletal muscle. *Am J Pathol* **154**, 591–600.
- Verdijk LB, Koopman R, Schaart G, Meijer K, Savelberg HHCM & van Loon LJC (2007). Satellite cell content is specifically reduced in type II skeletal muscle fibers in the elderly. *Am J Physiol Endocrinol Metab* **292**, E151-157.
- Verdijk LB, Snijders T, Drost M, Delhaas T, Kadi F & van Loon LJC (2014). Satellite cells in human skeletal muscle; from birth to old age. *Age (Dordr)* **36**, 545–557.
- Walro JM & Kucera J (1999). Why adult mammalian intrafusal and extrafusal fibers contain different myosin heavy-chain isoforms. *Trends in Neurosciences* **22**, 180–184.
- Walsh FS & Moore SE (1985). Expression of cell adhesion molecule, N-CAM, in diseases of adult human skeletal muscle. *Neurosci Lett* **59**, 73–78.
- Winter & Bornemann (1999). NCAM, vimentin and neonatal myosin heavy chain expression in human muscle diseases. *Neuropathology and Applied Neurobiology* **25**, 417–424.
- Wong A, Garcia SM, Tamaki S, Striedinger K, Barruet E, Hansen SL, Young DM & Pomerantz JH (2021). Satellite cell activation and retention of muscle regenerative potential after long-term denervation. *Stem Cells* **39**, 331–344.
- Xing H, Zhou M, Assinck P & Liu N (2015). Electrical stimulation influences satellite cell differentiation after sciatic nerve crush injury in rats: Satellite Cell Differentiation. *Muscle & Nerve* **51**, 400–411.
- Zampieri S et al. (2015). Lifelong physical exercise delays age-associated skeletal muscle decline. *J Gerontol A Biol Sci Med Sci* **70**, 163–173.

Dear Dr Soendenbroe,

Re: JP-RP-2022-282677R1 "Preserved stem cell content and innervation profile of elderly human skeletal muscle with lifelong recreational exercise" by Casper Soendenbroe, Christopher Lund Dahl, Christopher Meulengracht, Michal Tamáš, Rene B. Svensson, Peter Schjerling, Michael Kjaer, Jesper Løvind Andersen, and Abigail Louise Mackey

I am pleased to tell you that your paper has been accepted for publication in The Journal of Physiology.

NEW POLICY: In order to improve the transparency of its peer review process The Journal of Physiology publishes online as supporting information the peer review history of all articles accepted for publication. Readers will have access to decision letters, including all Editors' comments and referee reports, for each version of the manuscript and any author responses to peer review comments. Referees can decide whether or not they wish to be named on the peer review history document.

Are you on Twitter? Once your paper is online, why not share your achievement with your followers. Please tag The Journal (@jphysiol) in any tweets and we will share your accepted paper with our 23,000+ followers!

The last Word version of the paper submitted will be used by the Production Editors to prepare your proof. When this is ready you will receive an email containing a link to Wiley's Online Proofing System. The proof should be checked and corrected as quickly as possible.

Authors should note that it is too late at this point to offer corrections prior to proofing. The accepted version will be published online, ahead of the copy edited and typeset version being made available. Major corrections at proof stage, such as changes to figures, will be referred to the Reviewing Editor for approval before they can be incorporated. Only minor changes, such as to style and consistency, should be made a proof stage. Changes that need to be made after proof stage will usually require a formal correction notice.

All queries at proof stage should be sent to TJP@wiley.com

Yours sincerely,

Richard Carson
Senior Editor
The Journal of Physiology

P.S. - You can help your research get the attention it deserves! Check out Wiley's free Promotion Guide for best-practice recommendations for promoting your work at www.wileyauthors.com/eoo/guide. And learn more about Wiley Editing Services which offers professional video, design, and writing services to create shareable video abstracts, infographics, conference posters, lay summaries, and research news stories for your research at www.wileyauthors.com/eoo/promotion.

*** IMPORTANT NOTICE ABOUT OPEN ACCESS ***

Information about Open Access policies can be found here <https://physoc.onlinelibrary.wiley.com/hub/access-policies>

To assist authors whose funding agencies mandate public access to published research findings sooner than 12 months after publication The Journal of Physiology allows authors to pay an open access (OA) fee to have their papers made freely available immediately on publication.

You will receive an email from Wiley with details on how to register or log-in to Wiley Authors Services where you will be able to place an OnlineOpen order.

You can check if your funder or institution has a Wiley Open Access Account here <https://authorservices.wiley.com/author-resources/Journal-Authors/licensing-and-open-access/open-access/author-compliance-tool.html>

Your article will be made Open Access upon publication, or as soon as payment is received.

If you wish to put your paper on an OA website such as PMC or UKPMC or your institutional repository within 12 months of publication you must pay the open access fee, which covers the cost of publication.

OnlineOpen articles are deposited in PubMed Central (PMC) and PMC mirror sites. Authors of OnlineOpen articles are permitted to post the final, published PDF of their article on a website, institutional repository, or other free public server, immediately on publication.

Note to NIH-funded authors: The Journal of Physiology is published on PMC 12 months after publication, NIH-funded

authors DO NOT NEED to pay to publish and DO NOT NEED to post their accepted papers on PMC.

EDITOR COMMENTS

Reviewing Editor:

Both reviewers agree that you have satisfactorily responded to their comments, and this work is now suitable for publication in the Journal of Physiology. Congratulations.

REFEREE COMMENTS

Referee #1:

The authors have done an excellent job responding to my comments and I commend them for an important contribution to the field.

Referee #2:

Thank you for addressing my concerns. I believe this iteration of the manuscript is significantly improved. I have no further comments.

1st Confidential Review

01-Feb-2022